# Temporal control of human DNA replication licensing by CDK4/6-RB signalling and chemical genetics

Anastasia Sosenko Piscitello[1,5], Ann-Sofie Nilsson[2,5], Michael Hawgood [2,5], Abid H. Sayyid [2,3], Vasilis S. Dionellis[1], Giovanni Giglio [2], Bruno Urién [2], Pratikiran Bajgain [2], Sotirios G. Ntallis [1], Jiri Bartek [2,4] ✉, Thanos D. Halazonetis [1] ✉ & Bennie Lemmens [2] ✉

Cyclin-dependent kinases (CDKs) coordinate DNA replication and cell division, and play key roles in tissue homeostasis, genome stability and cancer development. The first step in replication is origin licensing, when minichromosome maintenance (MCM) helicases are loaded onto DNA by CDC6, CDT1 and the origin recognition complex (ORC). In yeast, origin licensing starts when CDK activity plummets in G1 phase, reinforcing the view that CDKs inhibit licensing. Here we show that, in human cells, CDK4/6 activity promotes origin licensing. By combining rapid protein degradation and time-resolved EdU-sequencing, we find that CDK4/6 activity acts epistatically to CDC6 and CDT1 in G1 phase and counteracts RB pocket proteins to promote origin licensing. Therapeutic CDK4/6 inhibitors block MCM and ORC6 loading, which we exploit to trigger mitosis with unreplicated DNA in p53-deficient cells. The CDK4/6-RB axis thus links replication licensing to proliferation, which has implications for human cell fate control and cancer therapy design.

To maintain healthy proliferating tissues, DNA replication needs to be coordinated with the molecular engines that drive cell division. A human cell must copy billions of DNA bases each cell division, which can only be accomplished in time by triggering DNA replication at thousands of start sites (origins) scattered across the human genome[1,2]. The activity of replication origins is defined by at least two distinct and temporally separated mechanisms: i) origin licensing and ii) origin firing (Fig. 1a). Origin licensing occurs in the G1 phase of the cell cycle and involves loading the replicative helicases onto DNA. Origin firing occurs in S phase and entails activation of the loaded helicases to allow bidirectional DNA synthesis. At the molecular level, origin licensing requires the origin recognition complex (ORC) to bind DNA sites across the genome, subsequent recruitment of CDC6 and CDT1 and loading of two minichromosome maintenance (MCM)

hexamer complexes to each replication origin. Origin firing requires Dbf4-dependent Cdc7 kinase (DDK) and cyclin-dependent kinases 1 or 2 (CDK1/2) to phosphorylate MCM double hexamers, which triggers the MCM helicases to unwind the DNA helix, recruit additional replisome components and initiate DNA synthesis (Fig. 1a)[1,2]. To prevent that certain parts of the genome are copied more than once, cells need to prevent origin re-licensing during S-phase. This is achieved by a number of partially redundant mechanisms, many of which require CDK activity[3–5]. Thus, in yeast, it is well-established that only after cell division, when overall CDK activity drops, origins can be licensed again, reinforcing a model where CDK activity is a central negative regulator of licensing. However, in yeast, a single CDK activity (Cdc28 in budding yeast or Cdc2 in fission yeast) is responsible for catalysing all major cell cycle transitions, while human cells have many different

[1]Department of Molecular and Cellular Biology, University of Geneva, Geneva, Switzerland. [2]Department of Medical Biochemistry and Biophysics, Karolinska Institutet, Science for Life Laboratory, Stockholm, Sweden. [3]Department of Civil, Environmental and Natural Resources Engineering, Luleå University of Technology, Luleå, Sweden. [4]Danish Cancer Institute, Copenhagen, Denmark. [5]These authors contributed equally: Anastasia Sosenko Piscitello, Ann-Sofie Nilsson, Michael Hawgood. ✉e-mail: jb@cancer.dk; thanos.halazonetis@unige.ch; bennie.lemmens@ki.se

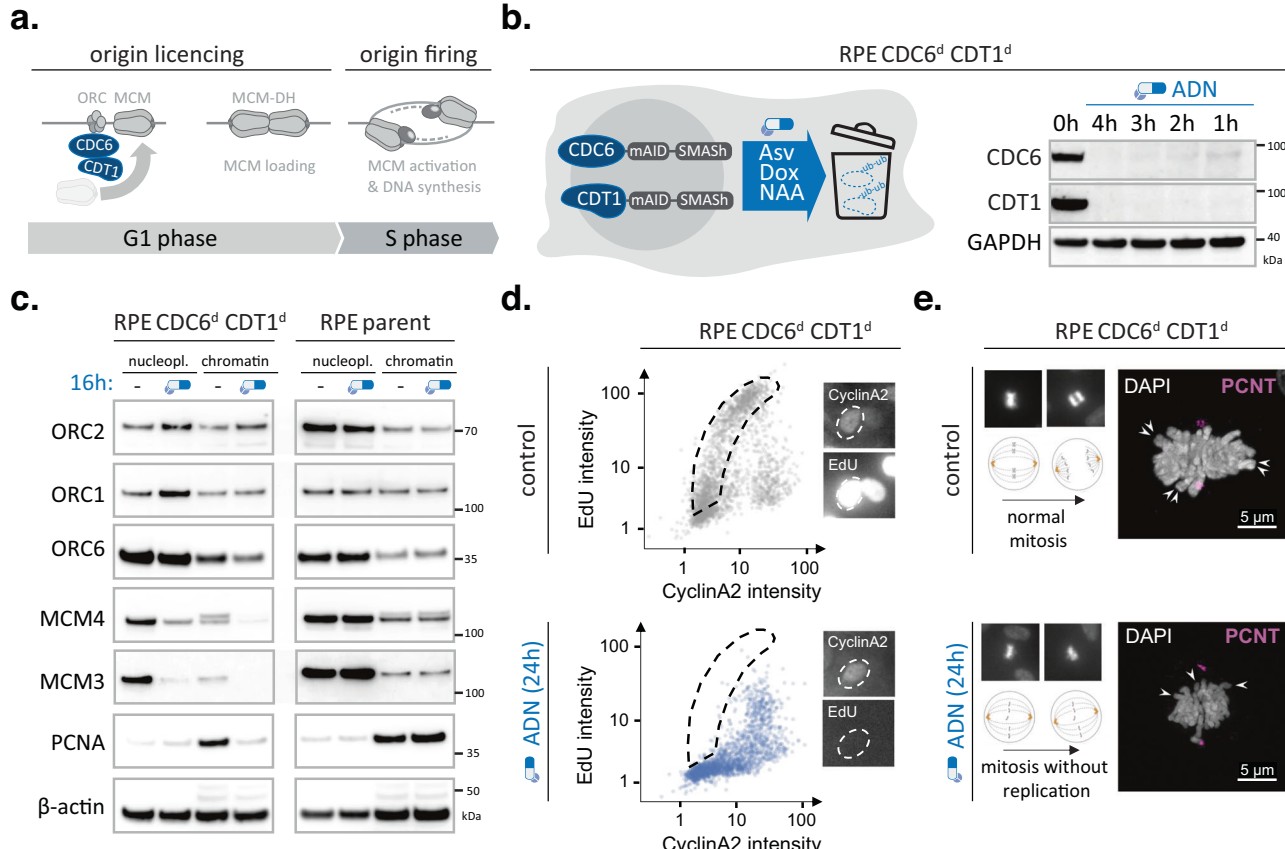

**Fig. 1 | CDC6 and CDT1 depletion abolishes DNA replication initiation while allowing cell cycle progression. a–e** Rapid degradation of CDC6 and CDT1 prevents MCM loading but not mitotic entry (**a**), overview of key DNA replication initiation steps, including CDC6 and CDT1-mediated loading of MCM double hexamers to license replication origins. **b** schematic overview of RPE CDC6d CDT1d cell model (left) and immunoblot validating rapid, ADN-induced depletion of endogenously tagged CDC6 and CDT1 (right). Western blot experiments were independently repeated twice with similar results. **c** Immunoblotting of nucleoplasmic and chromatin fractions of RPE CDC6d CDT1d cells and its parental RPE controls upon 16-h mock (hyphen) or ADN treatment (blue drug logo). Representative blots are shown, with the experiment repeated twice independently with consistent results. Source data are provided as a Source Data file. **d** QIBC analysis simultaneously monitoring DNA synthesis (EdU incorporation) and nuclear Cyclin A2 levels

in single RPE CDC6d CDT1d cells upon 24-h mock (grey) or ADN treatment (blue); Figure axes depict integrated fluorescence intensities (arb. units) per nucleus; >2300 cells per condition; dotted line highlights expected burst of DNA synthesis in early S-phase cells. Right panels depict representative IF images of early S phase cells, demonstrating undetectable EdU levels upon ADN treatment and near-saturated EdU signals upon mock treatment while using identical imaging conditions (**e**), mitotic cell fates of mock- or ADN-treated RPE CDC6d CDT1d cells, left panels illustrate representative DAPI images (upper panels) and schematic interpretation (bottom panels), while right panels depict representative confocal images of mitotic cells stained for Pericentrin (PCNT) and counterstained with DAPI; arrows indicate extruding paired sister chromatids (control condition) or single chromosomes (ADN condition); Scale bars, 5 μm.

cyclin-CDK complexes specialised for different cell cycle phases[6–8], and cyclin E-CDK2 has been shown to promote origin licensing in specific contexts[9], which prompted us to ask whether there are differences in how yeast and human cells regulate replication licensing.

Major players in controlling G1 phase progression and S phase commitment are the anaphase-promoting complex (APC/C), CDK1/2 and CDK4/6[10–12]. While selective targeting of APC/C or CDK1/2 have yielded limited clinical benefits (due to toxicity)[13–15], CDK4/6 inhibitors such as palbociclib have demonstrated remarkable efficacy against several breast cancer subtypes[16]. These results reignited the interest in therapeutic CDK inhibitors and indicated that CDK4/6 bear relevant activities that are different from other CDKs[17–19]. The retinoblastoma (RB) pocket protein family members RB1, RBL1, and RBL2 are the prime targets of CDK4/6, and their phosphorylation promotes cell cycle progression via E2F-dependent transcription programmes[12,17]. Pocket proteins also regulate chromatin dynamics independent of E2F and direct interactions between RB, cyclin D-CDK4, MCM7 and ORC1 complexes have implied functional connections between CDK4/6 signalling and origin licensing[20,21]. Defining if and how CDK4/6 controls replication, however, remains a challenge.

Here, we generated human cell models that allow direct control of origin licensing and applied selective CDK4/6 inhibitors, chromatin extraction assays and time-resolved EdU sequencing methods to identify a key role for the CDK4/6-RB axis in origin licensing during G1 phase. Combining CDK4/6 and origin licensing inhibition eliminates DNA replication initiation and triggers mitotic arrest in p53-deficient cells. We propose that the CDK4/6-RB axis coordinates cell cycle commitment with the build-up of DNA replication complexes, which differentiates replication initiation in yeast and humans.

## Results
### Turning off origin licensing within one cell cycle
We have previously developed protein depletion technologies to suppress initiation of DNA replication in human cells[22]. To fully inhibit DNA replication, we previously had to simultaneously block origin licensing (by CDC6 degradation) and origin firing (by CDC7 inhibition)[22]. Since licensing and firing are fundamentally distinct stages of replication initiation and CDC7 has possible confounding roles in checkpoint signalling[23], we generated a human cell model exclusively targeting origin licensing. To achieve high efficacy and

temporal resolution within a single G1 phase, our model combines two synergistic degron tags, SMASh and mAID, and expression of a recently identified co-factor ARF-PB1 that improves degradation dynamics[22,24]. The SMASh tag is a large self-cleaving peptide that rapidly removes itself from the protein of interest; upon addition of the small molecule Asunaprevir (ASV) self-cleavage is blocked, rendering newly translated fusion proteins unstable and/or dysfunctional (Supplementary Fig. 1a). At the same time, the mAID tag allows active recruitment of an ectopic E3-ligase (OsTIR) and upon addition of a small synthetic auxin molecule (NAA) triggers rapid ubiquitination and proteasomal degradation of the remaining target proteins (Supplementary Fig. 1a). To prevent constitutive degradation of mAID-tagged proteins and further improve protein degradation kinetics, we introduced a co-expression construct that allows doxycycline-(DOX) inducible expression of OsTIR and the PB1 domain of ARF16 (ARF-PB1) from the human Rosa26 safe harbour locus (Supplementary Fig. 1A). Using CRISPR-based genome-editing, we introduced this system into human RPE1 p53-/- cells and tagged both alleles of endogenous CDC6 and CDT1 with a mAID-SMASh double-degron. The resultant RPE Cdc6^d Cdt1^d cells showed rapid depletion of both target proteins - with levels of both full-length proteins being very low and undetectable at 1 and 4 h post-induction, respectively (Fig. 1b). CDC6 and CDT1 are both required for ORC to load the MCM helicase on DNA. Selective degradation of these proteins should, therefore, reduce MCM loading on chromatin without affecting ORC (Fig. 1a). Indeed, exposing RPE Cdc6^d Cdt1^d cell to the degron drugs ASV/DOX/NAA (ADN) for 16 h eliminated MCM4 and MCM3 from the chromatin fraction, while preserving ORC1 and ORC2 (Fig. 1c). Parallel treatments using the parental RPE1 p53-/- cells did not alter chromatin-bound MCM or ORC levels, verifying that the ADN treatment, by itself, did not compromise licensing (Fig. 1c).

To study how simultaneous depletion of CDC6 and CDT1 alters DNA replication and cell cycle progression, we performed quantitative image-based cytometry (QIBC) of large cell populations and for each cell plotted 5-Ethynyl-2′-deoxyuridine (EdU) nucleotide incorporation levels versus nuclear Cyclin A2 intensities. Asynchronous, mock-treated RPE Cdc6^d Cdt1^d cells showed a typical arc-shaped distribution, indicative of efficient DNA replication initiation upon S-phase entry when Cyclin A2 levels start to accumulate (Fig. 1d). In contrast, ADN-treated RPE Cdc6^d Cdt1^d cells failed to trigger high EdU incorporation rates in early S phase, while allowing CyclinA2 accumulation, revealing a strong and selective defect in DNA replication initiation in the absence of CDC6 and CDT1 (Fig. 1d). Independent FACS experiments confirmed a complete block of DNA replication upon ADN treatment, which relied on degron-tagged CDC6 and CDT1 (Supplementary Fig. 1b). Notably, depleting CDC6 and CDT1 prevents the production of DNA replication forks, and thus is fundamentally different from stopping DNA replication by impeding fork progression, which is known to cause DNA replication stress, DNA damage checkpoint activation and cell cycle arrest[25]. Directly comparing degradation of CDC6 and CDT1 with classical fork stalling agents such as hydroxyurea (HU) indicates that i) the RPE Cdc6^d Cdt1^d cells are proficient in DNA damage signalling and ii) loss of origin licensing does not trigger DNA replication stress markers such as H2AX or RPA hyperphosphorylation (Supplementary Fig. 2a). In fact, depletion of CDC6 and CDT1 suppressed spontaneous and HU-induced DNA damage, as expected in the absence of replication intermediates (Supplementary Fig. 2a). Moreover, since it did not prevent Cyclin A2 accumulation and mitotic entry, it led to mitotic cells without apparent sister-chromatids (Fig. 1e, Supplementary Fig. 1c and Supplementary Fig. 2b).

The ability to rapidly deplete CDC6 and CDT1 allowed us to ask whether origin licensing can occur any time during the G1 phase of the cell cycle. RPE Cdc6^d Cdt1^d cells, treated with nocodazole, were collected by mitotic shake-off, plated in fresh media and allowed to proceed through G1 into S phase. With this protocol, the cells entered S phase between 12 and 14 h after mitotic exit (Fig. 2a). To determine

when origins are licensed during G1, the ADN drug mixture was added to the cells either during mitosis and removed at various time points in G1 or was added at various time points in G1 and kept until the 14 h time point after mitotic exit, at which time the fraction of EdU-positive cells was determined by flow cytometry. ADN reduced the number of EdU-positive cells in a treatment duration-dependent manner and, independent of whether the cells were exposed to ADN in the first or second half of G1 (Fig. 2b). These results suggest that origin licensing can occur in a cumulative manner throughout the G1 phase, consistent with previous reports[26,27].

While restricting origin licensing to specific G1 time windows did not prevent entry into S phase, the effect on the firing of individual origins could not be determined by the flow cytometry analysis. Moreover, the flow cytometry analysis classified cells as EdU-positive or EdU-negative but did not provide a quantitative assessment of the level of DNA replication initiation. We, therefore, resorted to a more detailed analysis of initiation of DNA replication by employing EdUseq, a method that can map origin firing in a genome-wide manner[26,28]. In line with the flow cytometry data, active degradation of CDC6 and CDT1 for 12 h throughout G1 (treatment M→12) abolished EdU incorporation at origins, as seen by inspecting a representative genomic region (chromosome 7, Fig. 2c) or by plotting the average EdUseq signal of the thousand most efficient origins (Fig. 2d).

Shorter treatments with ADN resulted in partial suppression of origin firing with the magnitude of the effect correlating with the length of ADN treatment. Thus, ADN treatments during the first 4 or 8 h of G1 phase (treatments M→4 and M→8) led to 85% and 94% reductions in average origin activity, respectively (Fig. 2d, e). Similarly, ADN treatments during the last 4 or 8 h of G1 phase (treatments 8→S and 4→S) led to 60% and 78% reductions in average origin activity, respectively (Fig. 2d, e). These results further indicate that ADN treatments suppressed origin firing when given either at the beginning or end of G1 phase.

To study if certain origins are licensed in early versus late G1 phase, we compared for each origin its EdUseq signal in the cells treated with ADN for the first 4 h of G1 (treatment M→4) versus the cells treated with ADN for the last 8 h of G1 (treatment 4→S). The comparisons demonstrated a highly significant correlation of signal intensities ($R^2 = 0.78$) indicating that the majority of origins are impacted similarly by the two ADN treatments (Fig. 2e, f). A similar conclusion can be reached by examining the EdUseq origin firing data for all the other ADN treatments described above (Supplementary Fig. 3). Together, these data indicate that efficient DNA replication initiation requires MCM loading activities throughout G1 phase and that there is no preference for specific origins to be licensed early versus late in the G1 phase of the cell cycle.

## CDK4/6 inhibition stalls origin licensing through RB

Given the key role of CDK4/6 activity in G1/S phase transition, and the fact that the expression profile of its activating partner Cyclin D1 mirrors the timing of MCM loading[27,29–31], we wondered if CDK4/6 activity could be the missing link between replication licensing and cell cycle control. Selective CDK4/6 inhibitors cause a potent, yet fully reversible, G1 phase arrest, and thus are extensively used to obtain synchronous cell cultures[32,33]. Alternatively, the plant amino acid mimosine can be used to cause a reversible G1/S phase arrest[34,35]. Mimosine effectively stalls human cells before the onset of DNA replication, hence creating a synchronised population with maximum levels of loaded MCM (Supplementary Fig. 4a). To test if CDK4/6 activity is required for efficient origin licensing, we subjected RPE1^p53-/- cells to mimosine and/or two independent CDK4/6 inhibitors, palbociclib or Trilaciclib[17,34] and compared chromatin-associated MCM2 and CDC45 levels in fractionated lysates (Fig. 3a). Treatment with palbociclib or Trilaciclib diminished the levels of chromatin-associated MCM2 compared to mimosine or mock-treated controls. Adding

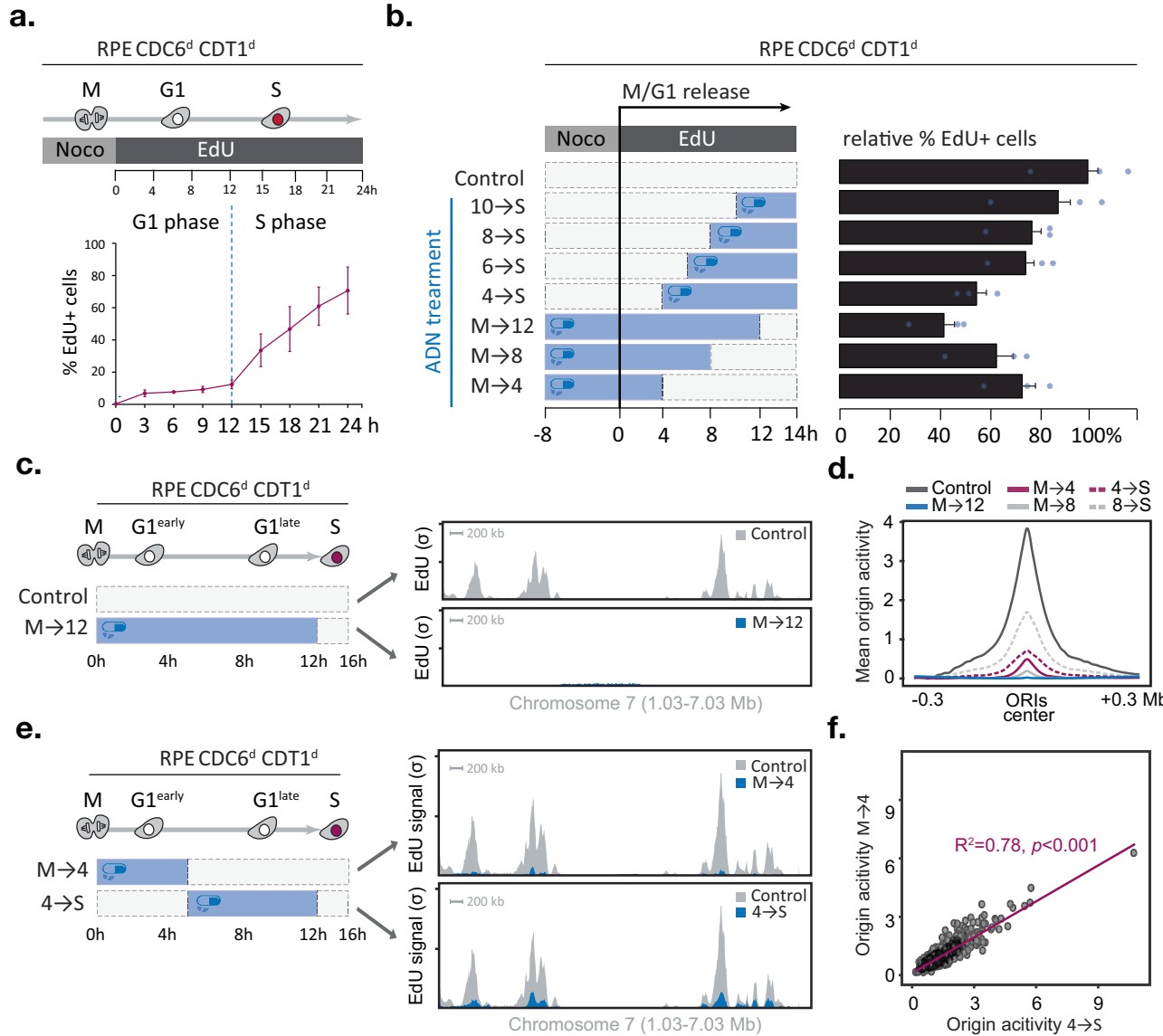

**Fig. 2 | CDC6 and CDT1 are needed throughout G1 phase to establish origin activity genome-wide. a–f** CDC6 and CDT1 are required in early and late G1 phase to promote DNA replication initiation genome-wide, **a** Experimental outline (top) and line graph (bottom) depicting the mean percentage of EdU-positive cells after mitotic release determined by FACS ($n = 3$ technical replicates, error bars indicate SD). **b** Experimental outline (left) and bar graph (right) depicting the mean percentage of EdU-positive cells 14 h after mitotic release for each condition ($n = 3$ technical replicates, error bars indicate SD). CDC6 and CDT1 are needed in early and late G1 phase to promote origin activity; **c** Outline of the experiment and corresponding replication initiation profiles (EdUseq-HU) at a representative genomic region in mock (grey) or ADN-treated (blue) cell populations and collected 16 h after mitotic shake-off. Bin resolution, 10 kb; scale bar, 200 kb; $\sigma$, sigma (normalised number of sequence reads per bin divided by its SD), lower tick

$\sigma = 100$, higher tick $\sigma = 200$. **d** Average origin activity (i.e. mean $\sigma$ values) at 1 Kb resolution around 1000 predefined, most active early S-phase origins after different ADN treatments (as outlined in **b**), **e** Outline of the experiment and corresponding replication initiation profiles (EdUseq-HU) at a representative genomic region in mock (grey) or ADN-treated (blue) cell populations and collected 16 h after mitotic shake-off. Bin resolution, 10 kb; scale bar, 200 kb; $\sigma$, sigma (normalised number of sequence reads per bin divided by its SD), lower tick $\sigma = 100$, higher tick $\sigma = 200$. **f** scatter plot comparing EdUseq-HU ($\sigma$) values at 1000 individual early S-phase origins after treatment M→4 or 4→S (as outlined in **e**). Linear regression fit (purple line) with coefficient of determination ($R^2$) indicating the proportion of variance explained between the two datasets; slope significance determined using a two-tailed t-test. No adjustment for multiple comparisons was performed. Source data are provided as a Source Data file.

palbociclib prior to mimosine reduced the levels of chromatin-associated MCM2 compared to mimosine-only controls, suggesting that CDK4/6 inhibition arrested cells in a distinct G1 phase state with incomplete origin licensing (Fig. 3a). Indeed, palbociclib or Trilaciclib arrested cells with increased Cyclin D1 and reduced Cyclin A2 levels, indicative of a G1 phase arrest prior to APC/C inactivation, while mimosine treatment caused elevated Cyclin A2 and reduced Cyclin D1 levels, indicative of an arrest at the G1/S phase transition post APC/C inactivation (Fig. 3a and[11,34]). Notably, the changes in chromatin-associated MCM2 are paralleled with changes in RB phosphorylation in

the nucleoplasm fractions, substantiating a link between CDK4/6 activity and MCM loading efficacy (Fig. 3a). Follow-up studies using p53-proficient and p53-deficient epithelial cells, as well as primary human fibroblasts, confirmed that palbociclib limits MCM loading and indicated that this defect does not rely on p53 status (Supplementary Fig. 4b). Moreover, independent high-content imaging data confirmed a 4-5-fold decrease in chromatin-associated MCM2 and MCM6 in G1 cells upon palbociclib addition (Fig. 3b).

Due to its selective and reversible nature, CDK4/6 inhibition has become a widespread and recommended approach to synchronise

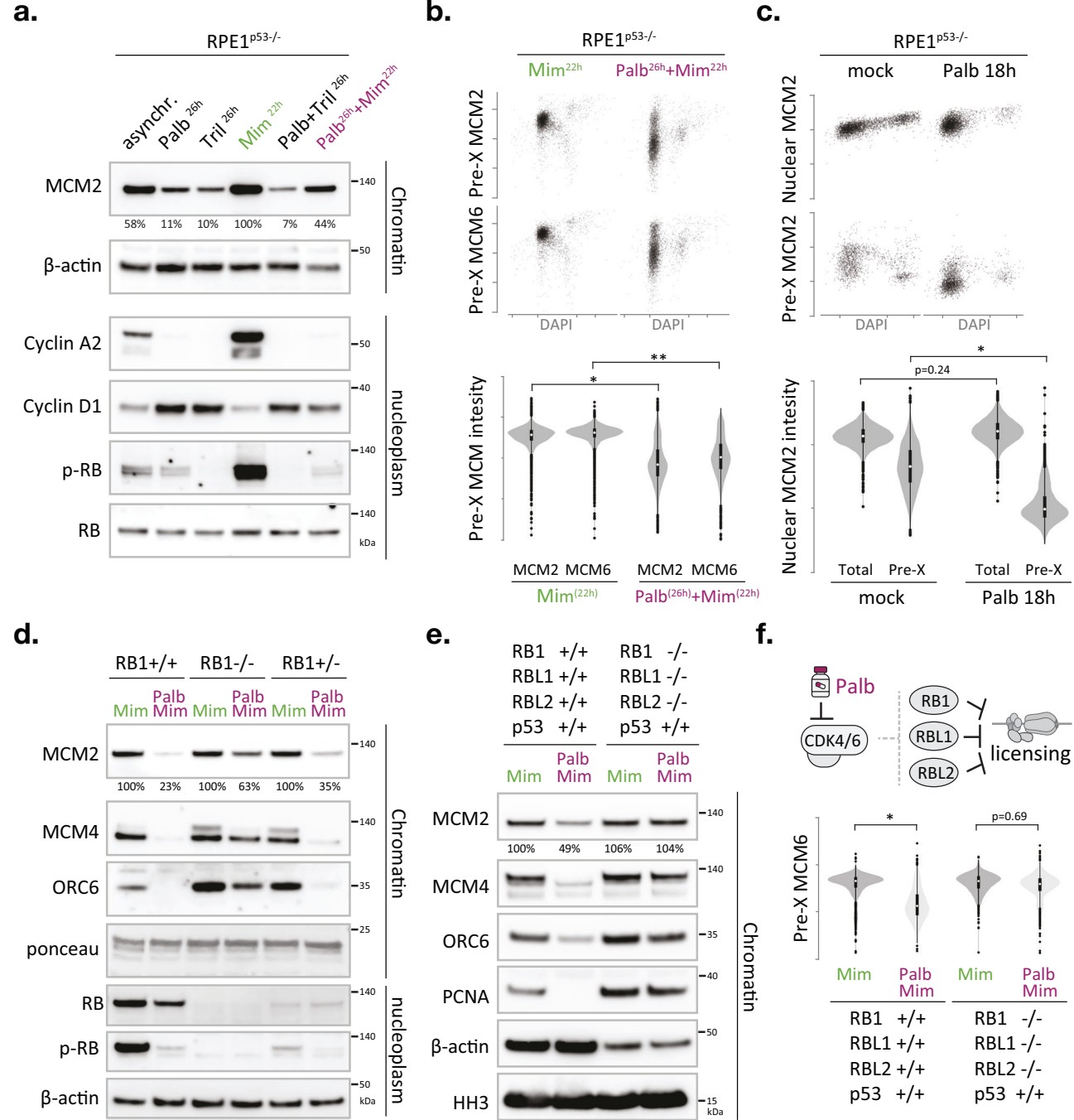

cells[32,33], however, long-term CDK4/6 inhibition (>3 days) should be prevented as it causes MCM complex instability, asynchronous releases and DNA damage[36]. We indeed find that long-term palbociclib and mimosine treatments (4–7 days) cause a stark drop in MCM2 levels in total cell lysates, while short treatments (1–2 days) do not significantly alter total MCM2 levels (Supplementary Fig. 4c). Adding the proteasome inhibitor MG132 to palbociclib and mimosine treated cells increases CDC6 and CDT1 expression but does not alleviate the MCM loading defect, suggesting that CDC6 or CDT1 protein levels are not limiting in this context and targeted MCM degradation is not the leading cause of the observed licensing defect (Supplementary Fig. 4d).

To directly compare the immediate effects of CDK4/6 inhibition on total and chromatin-bound MCM proteins (independent of mimosine), we exposed cells to palbociclib (200 nM) for 18 h and performed high-content imaging on pre-extracted and non-pre-extracted cells

and quantified integrated MCM2 intensities in individual G1 phase cells. While 18 h palbociclib treatment did not significantly affect the mean intensity of total nuclear MCM2, it reduced chromatin-bound MCM2 more than 4-fold (Fig. 3c). Independent experiments in U2OS and RPE1 cells confirmed that a short pulse of Palbociclib (200 nM, 8-h) is sufficient to reduce chromatin-bound MCM2 levels in G1 phase nuclei (Supplementary Fig. 4e). We thus conclude that CDK4/6 inhibition causes a potent and immediate stall in origin licensing, which occurs before MCM protein levels become limiting.

CDK4/6 regulates cell cycle commitment by phosphorylating the RB family of pocket proteins, which modulates their binding to the E2F transcription factors that drive G1/S phase transition[17]. Interestingly, RB proteins bind multiple DNA replication initiation factors, such as ORC1[37], MCM7[38] and BRD4[21]. Previous in vitro studies have indicated that human CDK activities evict RB from the ORC1-CDC6 complex while preserving the interaction between ORC1 and CDC6[39]. To test if

**Fig. 3 | CDK4/6 inhibitors prevent the completion of origin licensing via the RB protein family. a–f** The CDK4/6-RB axis controls origin licensing proficiency. **a** Immunoblot monitoring the effect of CDK4/6 inhibitors and/or mimosine on chromatin-bound MCM2 levels and corresponding cell cycle markers in nucleoplasm fractions. RB total and β-actin served as loading control (see Supplementary Fig. 4b) Representative blots are shown, with the experiment repeated twice. Values under immunoblots represent relative percentage of MCM2 signal compared to the Mimosine positive control. **b** QIBC analysis monitoring chromatin-bound MCM2 or MCM6 levels in single cells, relative to DNA content (DAPI), after expose to mimosine only or palbococlib and mimosine (analogous to the treatments in **a**). Scatterplots show MCM2/MCM6 levels in >1000 pre-extracted nuclei per condition. Violin plots depict distribution of MCM2/MCM6 signals in pre-extracted G1 phase nuclei (classified based on DAPI). The box shows the interquartile range (IQR) from 25th (Q1) to 75th (Q3) percentile; the white dot in the box corresponds to the median. Whiskers extend to the minimum and maximum values within 1.5 × IQR from Q1 and Q3, and dots outside this range represent potential outliers. Asterisks indicate a significant disparity between population means; *$p = 0.0007$; ** $p = 0.0003$; two-tailed paired t test; $n = 4$ technical replicates. **c** QIBC analysis monitoring total nuclear or chromatin-bound MCM2 levels in single cells, relative to DNA content (DAPI), after palbococlib expose. Scatterplots show MCM2 levels in >1000 nuclei per condition. Violin plots depict distribution of MCM2 signals in G1 phase nuclei (classified based on DAPI). The box shows the interquartile range (IQR) from 25th (Q1) to 75th (Q3) percentile; the white dot in the box corresponds to the median. Whiskers extend to the minimum and maximum values within 1.5 × IQR from Q1 and Q3, and dots outside

this range represent potential outliers. Asterisk indicates a significant disparity between population means, $p = 0.023$; two-tailed paired t test; $n = 4$ technical replicates. **d** Immunoblot examining the effect of RB1 deficiency on palbociclib-induced licensing defects. The changes in chromatin-bound MCM2 are mirrored by chromatin-bound MCM4 and ORC6. Total RB protein and RB (S807/811) phosphorylation status is verified in nucleoplasm fractions. Ponceau S staining of bulk histones (10–25 kDa) and immunodetection of β-actin served as loading control. Nucleoplasmic CDC6 and CDT1 levels are shown in Supplementary Fig. 6a. **e** Immunoblot monitoring the effect of palbociclib and/or mimosine on origin licensing in RPE cells lacking all RB pocket proteins. **d**, **e** Cells were treated as in Fig. 4b and Supplementary Fig. 4b. Nucleoplasmic CDC6, CDT1 and RB phospho-RB (S807/811) phosphorylation levels are shown in Supplementary Fig. 6b. **f** QIBC analysis monitoring chromatin-bound MCM6 levels in single cells treated as in (**e**). Scatterplots show MCM6 levels in >1000 pre-extracted nuclei per condition. Violin plots depict distribution of MCM6 signals in pre-extracted G1 phase nuclei (classified based on DAPI). The box shows the interquartile range (IQR) from 25th (Q1) to 75th (Q3) percentile; the white dot in the box corresponds to the median. Whiskers extend to the minimum and maximum values within 1.5 × IQR from Q1 and Q3, and dots outside this range represent potential outliers. Asterisk indicates a disparity between population means; $n = 5$ technical replicates per condition, $p = 0.00001$ (two-tailed paired t test). Asynchr. asynchronous, Palb palbociclib, Tril Trilaciclub, Mim Mimosine, pre-X preextracted/chromatin bound. Mimosine only and Minosine+Palbociclib treatments are highlighted with green and purple labels, respectively, and performed as depicted in Supplementary Fig. 4b. Source data are provided as a Source Data file.

CDK4/6 inhibitors block licensing via RB, we obtained isogenic RPE1 clones[40] with either normal levels of RB, no detectable RB or severely reduced levels of RB, and exposed these clones to palbociclib and mimosine, or mimosine alone, and determined the relative amount of chromatin-bound MCM2 and MCM4. RB deficiency alleviated the MCM loading defect imposed by palbociclib, directly implicating the tumour suppressor RB in origin licensing regulation (Fig. 3d and Supplementary Fig. 5a). The fact that the rescue is not complete and a relatively low dose of palbociclib (200 nM) can still reduce the level of chromatin-bound MCM in the absence of RB indicates that additional CDK4/6 targets exist that limit MCM loading in human cells. Humans have two additional RB-like proteins, RBL1/p107 and RBL2/p130, and overexpression of the latter is reported to impede DNA synthesis in a *Xenopus* in vitro replication assay[38]. We therefore investigated the effect of palbociclib in human RPE1 cells deficient for all three RB genes and found that the palbociclib-induced MCM loading defect is completely restored in RB1/RBL1/RBL2 triple knockout cells[41] (Fig. 3e and Supplementary Fig. 5b). These experiments also revealed that palbociclib impaired ORC6 recruitment in an RB-dependent manner, supporting a key upstream role for CDK4/6 in origin licensing (Fig. 3d-e).

Given that RB loss is a well-established mechanism of resistance to CDK4/6 inhibitors in breast cancer patients, we wished to verify these findings using independent breast cancer models. A direct comparison of an RB-deficient (HCC1937) and two RB-proficient (MCF7 and MDA-MB-231) epithelial breast cancer cell lines substantiated the close correlation between RB (S807/811) phosphorylation and origin licensing efficacy (Supplementary Fig. 5c). These data also confirmed that Palbociclib effectively stalls the chromatin-recruitment of MCM2, MCM4 and ORC6 in RB-proficient cells and that this effect is alleviated in RB-deficient cells (Supplementary Fig. 5c).

Recent efforts to tackle therapy resistance involve proteolysis targeting chimeras (PROTACs)—small molecule drugs that trigger selective degradation of proteins in vivo. Notably, PROTACs degrading CDK4/6 have shown greater efficacy than traditional CDK4/6 kinase inhibitors in RB1-deficient cancer models, yet the mechanism behind this increased effectiveness remains to be clarified[42,43]. We find selective PROTACs targeting CDK4 (BSJ-04-132) or CDK6 (BSJ-03-123) to arrest human RPE1 cells in an under-licensed state (Supplementary Fig. 6a-d). While combined CDK4 and CDK6 PROTAC treatment and Palbociclib treatment were equally effective in stalling MCM loading

(Supplementary Fig. 6c-d), we observed several notable differences at the molecular level. First, while the PROTACs reduced CDK4/6 protein levels, the catalytic inhibitor Palbociclib increased CDK4/6 protein levels (Supplementary Fig. 6a). Second, Palbociclib inhibited RB phosphorylation more effectively than the combined treatment with CDK4 and CDK6 PROTACs, even when the PROTACs were used at higher concentrations to compensate for potential reductions in catalytic site activity (Supplementary Fig. 6a)[43]. These findings support a model in which (i) both CDK4 and CDK6 promote cell cycle progression and MCM loading in human cells, and (ii) the redundant CDK activities phosphorylate RB in absence of CDK4/6 proteins. These results also confirm the strong concordance between S/G2 phase entry (Supplementary Fig. 6b) and origin licensing proficiency (Supplementary Fig. 6c-d), indicating molecular coupling between cell cycle and DNA replication commitment. We propose that CDK4/6 activity counteracts pocket proteins in G1 phase to coordinate origin licensing and cell cycle commitment.

## RB blocks origin licensing independent of E2F-driven CDC6, CDT1 or MCM6 expression

A key role of CDK4/6-RB axis is to regulate E2F-dependent transcription[17]. Because CDC6, CDT1, and MCM6 are considered direct E2F-targets[44], we investigated whether the expression of these proteins becomes limiting following treatment with Palbociclib. While we found nuclear CDC6 and CDT1 levels to be higher in RB knockout cells compared to their RB-proficient controls (as predicted for E2F target genes), we noted that the Palbociclib-induced changes in nuclear CDT1 and CDC6 expression did not correlate with the changes in origin licensing (Fig. 3d-e and Supplementary Fig. 5a-c). We found Palbociclib treatment to increase nuclear CDT1 levels while reducing licensing, suggesting that transcriptional control of CDT1 is not limiting in this context. While Palbociclib treatment reduced nuclear CDC6 levels, it did so in an RB-independent manner (Fig. 3d-e and Supplementary Fig. 5a-c), suggesting that the effect of CDK4/6 activity on CDC6 expression/protein stability and its effect on MCM loading (which is RB-dependent) can be uncoupled.

Earlier work by the Meyer and Diffley laboratories indicated that human CDK2 activity is needed in G1 phase to promote the transcription and protein stability of CDC6[9,12]. To test if CDC6 expression is limiting for origin licensing upon CDK4/6 inhibition, we made use of a

human bronchial epithelial cell model that allows inducible, E2F-independent expression of CDC6 (HBEC CDC6 Tet-ON)[45]. While doxycycline treatment caused significant CDC6 overexpression in all cell cycle stages (Supplementary Fig. 7a), it did not rescue the palbociclib-dependent origin licensing defect in HBEC CDC6 Tet-ON cells (Supplementary Fig. 7b). QIBC and biochemical analysis confirmed that palbociclib impaired MCM2, MCM4 and ORC6 loading (Supplementary Fig. 7b, c). Notably, CDC6 chromatin occupancy was not impaired by CDK4/6 inhibition, revealing that human CDKs regulate origin licensing post CDC6 recruitment and beyond CDC6 expression and/or protein stability (Supplementary Fig. 7d). To determine whether CDT1 levels could be limiting in this context, we transfected HBEC CDC6 Tet-ON cell with a human CDT1 overexpression construct and found that elevated CDT1 levels did not restore MCM2 loading in cells treated with palbociclib and doxycycline (Supplementary Fig. 7e). These observations are in line with origin licensing not being restored when CDT1 and CDC6 are stabilised by proteasome inhibition (Supplementary Fig. 4d) and the fact that nuclear CDT1 levels do not correlate with impaired origin licensing upon CDK4/6 inhibition (Supplementary Fig. 5a-c). We conclude that short-term CDK4/6 inhibition can block origin licensing in cells where expression of the MCM loaders CDC6 and CDT1 is not limiting.

Since MCM6 is a validated E2F target with multiple E2F binding elements in its promoter[46], we directly compared total and chromatin-bound MCM6 levels upon Palbociclib exposure in RPE1 cells. QIBC analysis revealed that 18 h palbociclib treatment caused a stark reduction in chromatin-bound MCM6 in G1 phase cells without changing the total levels of MCM6 in G1 nuclei (Supplementary Fig. 5d). Parallel quantifications of MCM2 in these cells revealed identical results (Supplementary Fig. 5d). To further study the impact of the RB-E2F axis in licensing control, we transiently complemented RB1-deficient RPE1 cells with wildtype RB (RB^wt) or mutant RB defective in E2F binding (RB[661W])[47] and found that both RB constructs significantly stalled MCM loading in Palbociclib-treated G1 cells (Supplementary Fig. 8a, b), further supporting a model where RB pocket proteins block origin licensing independent of E2F-driven transcriptional regulation. In *Xenopus* extracts, Rb and p130 impede DNA synthesis via a direct interaction with the C-terminal domain of MCM7 (MCM7-CT)[38]. We find that ectopic expression of MCM7-CT in human RPE1 cells prevents both RB phosphorylation and chromatin-recruitment of MCM2, substantiating a functional link between RB regulation and origin licensing (Supplementary Fig. 9).

## CDK4/6 inhibitor-induced licensing defects can be perpetuated to cause mitosis with unreplicated DNA

Human cells license many more origins in G1 phase than are fired in an unperturbed S phase[48], so we wondered if the observed reduction in ORC6 and MCM loading upon CDK4/6 inhibition also affected the pool of origins required for genome replication. The non-linear correlation between MCM detection and origin activity as well as the fact that CDK4/6 controls S-phase entry, make this a challenging question to address, but our biochemical analysis showed encouraging signs that the relevant pool of MCM complexes was affected. While RB deficiency partially restored MCM4 loading in cells treated with mimosine and palbociclib, we noted an apparent lack of the upshifted MCM4 band compared to the mimosine-only control, implying that CDK4/6 inhibition alters the nature of the loaded MCM complexes and/or their interaction with DDK (Fig. 3d). Recent cryo-EM studies have demonstrated that DDK requires MCM complexes to be in a double-hexameric state to be able to phosphorylate MCM2 and MCM4, which is the key step towards origin firing[49–52]. If CDK4/6 activity would be needed to load the pool of MCM complexes that drive DNA synthesis and act upstream of the MCM loading factors CDC6 and CDT1, one would expect cells to be sensitive to CDC6 and CDT1 depletion upon CDK4/6 inhibitor release. To test this

hypothesis, we made use of our RPE Cdc6^d Cdt1^d cells and devised a quantitative IF setup that allowed us to rapidly block MCM loading upon palbociclib release and study cell cycle progression and DNA replication at single-cell resolution (Fig. 4a). If licensing did not occur during the transient palbociclib arrest, subsequent loss of CDC6 and CDT1 should sustain the licensing defect but allow S-phase entry. In line with this hypothesis, we found that palbociclib release in the presence of CDC6 and CDT1 led to efficient DNA replication, while palbociclib release upon CDC6 and CDT1 degradation led to a major drop in DNA synthesis (Fig. 4b, c). Combining palbociclib release with CDC6 and CDT1 degradation diminished EdU incorporation to undetectable levels in Cyclin A2-positive mid-S-phase cells, demonstrating that the majority of origins in CDK4/6 inhibited cells still require MCM loading activities in order to drive genome-wide DNA synthesis. Notably, the lack of EdU incorporation was not due to a failed drug release/persistent G1 arrest, since the CDC6 and CDT1-depleted cells did accumulate Cyclin A2 and ultimately entered mitosis (Fig. 4b, d). The proficiency in mitotic entry allowed us to independently confirm the failure to initiate DNA replication, as cytological analysis revealed abundant small DAPI-stained metaphase plates and a significant reduction in anaphase figures in cells depleted of CDC6 and CDT1 upon palbociclib release (Fig. 4d). Analogous assays using RL5a, a small molecule hampering ORC-DNA interactions[53], confirmed these results and indicated that palbociclib impairs replication independently of our Cdc6^d Cdt1^d model and that combined CDK4/6 and origin licensing inhibition can be used to effectively trigger aberrant mitosis in p53-deficient cells (Supplementary Fig. 10a, b). The same palbociclib/RL5a combination treatments triggered a potent interphase arrest in p53-proficient cells, confirming that human RPE1 cells have a functional origin licensing checkpoint[54,55]. Biochemical analysis of the p53-deficient cells validated the uncoupling of DNA replication and cell cycle progression, as CDC6 and CDT1 depletion upon palbociclib release reduced chromatin-bound PCNA despite increasing levels of Cyclin A2 (Fig. 4e). Directly comparing the chromatin states after palbociclib release, with or without CDC6 and CDT1 degradation, revealed a scenario where i) ORC1 levels are elevated, suggestive of failed ORC exclusion, and ii) phosphorylation of MCM4 and MCM2 (Ser 53) is impaired, which are independent indicators of failed MCM double-hexamer formation (Fig. 4e). Degradation of CDC6 and CDT1 upon palbociclib release did not reduce the level of chromatin-bound CDC7 nor did it impair MCM2 phosphorylation on an CDC7-independent site (Ser27)[56], suggesting that both the MCM substrate and CDC7 kinase are available, yet their functional interaction is blocked[49–51,57]. To test if these chromatin changes are not a general response to CDC6 or CDT1 degradation, we performed an analogous assay upon mimosine release, which revealed no difference in ORC1, PCNA or MCM phosphorylation status despite efficient CDC6 and CDT1 depletion (Supplementary Fig. 10c, d). Finally, replacing palbociclib with an independent CDK4/6 inhibitor, Trilaciclib, confirmed a significant reduction in DNA synthesis upon CDC6 and CDT1 depletion (Fig. 4f). Together, these data support a model where CDK4/6 activity controls a relevant pool of replication origins and acts prior to S-phase to generate productive MCM complexes on chromatin.

## CDK4/6 activity is required at the time of origin licensing

We next set out to establish when CDK4/6 activity is required during G1 phase and how relatively short pulses of CDK4/6 inhibition affect MCM loading and origin activity genome-wide. Previous live-cell imaging studies in CHO and MCF7 cells detected stable MCM loading throughout G1 phase but also found dynamic chromatin-MCM interactions to occur as early as telophase, implying that the first steps of origin licensing start already at mitotic exit[58,59]. When we exposed asynchronous RPE1 cells to a short 8-h pulse of Palbociclib, we

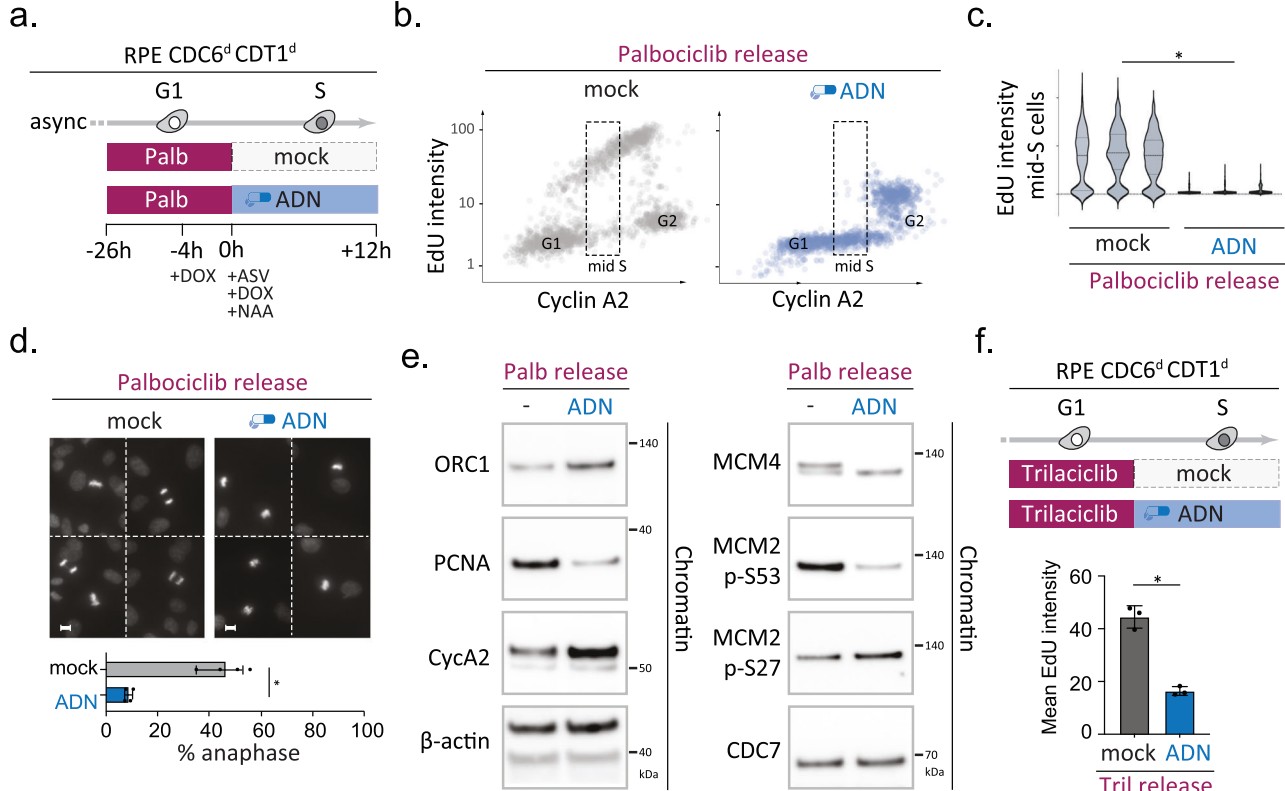

**Fig. 4 | CDK4/6-inhibition causes an origin licensing defect upstream of CDC6/CDT1 function, which can be sustained to trigger replication failure and aberrant mitosis in p53-deficient cells. a–f** Sequential CDK4/6 and licensing inhibition triggers mitosis with unreplicated DNA. **a** outline of the experiment. **b** QIBC analysis simultaneously monitoring DNA synthesis (EdU incorporation) and nuclear Cyclin A2 levels in single RPE CDC6d CDT1d cells 12 h after palbociclib release (>1000 cells per condition). Figure axes depict integrated fluorescence intensities (arb. units) per nucleus; Upon release cells were either mock treated (grey) and exposed to ADN treatment (blue); dotted line box highlights mid S-phase cells (classified based on CyclinA2). **c** violin plots depict distribution of EdU signals in three independent mid S-phase populations per condition, treated and classified as in (**a, b**) (*n* = 300 mid-S phase cells per replicate). Asterisk indicates a significant disparity between population means; *n* = 3 technical replicates per condition, *p* = 0.00002 (two-tailed paired t test). **d** mitotic phenotypes upon treatment described in (**a**); compound DAPI images show representative mitotic

nuclei; scale bars, 10 μm; bar graph depicts quantification of the percentage of anaphase nuclei among mitotic cells; bars indicate mean % of anaphase-like nuclei; error bars indicate SD of four independent replicates. Asterisk indicates *p* = 0.0002 (two-tailed paired t test). **e** Immunoblots examining the chromatin occupancy of indicated proteins and/or phosphorylated epitopes upon treatment described in (**a**); the reduction of PCNA as well as MCM4 and MCM2 serine 53 phosphoryation confirm a licensing defect. β-actin served as loading control. Representative blots are shown, with the experiment repeated twice independently with consistent results. **f** Top panel depicts outline of the experiment and bar graph below shows mean integrated intensity of nuclear EdU signals for each condition (arb. units); error bars indicate SEM of three independent experiments; black dots indicate replicate means. Asterisk indicates a significant disparity between population means; *p* = 0.0004 (two-tailed paired *t* test). Source data are provided as a Source Data file.

detected a significant loss of MCM2 intensity in pre-extracted G1 phase nuclei (Supplementary Fig. 4e), indicating that non-transformed RPE1 cells require CDK4/6 activity during G1 phase or a few hours before (i.e. G2/M phase) for effective and stable MCM loading.

Under unchallenged conditions, the majority of G1 phase RPE1 cells are in a high MCM state while only a few are in a low MCM state, reflecting efficient MCM2 loading upon mitotic exit (Supplementary Fig. 11a, b). Upon a short 8-h pulse of Palbociclib, many cells remain stuck at the low MCM state and only a minority of RPE1 cells reached the high MCM state (Supplementary Fig. 11c). To study the role of CDK4/6 activity in G2/M versus G1 phase, we synchronised RPE1 cells by mitotic shake-off, either after a 3-h nocodazole pulse or a 3-h pulse with nocodazole and Palbociclib. Mitotic cells were released in the absence or presence of Palbociclib and 8 h later fixed and analysed for chromatin-bound MCM2 by QIBC (Supplementary Fig. 11d). Importantly, inhibiting CDK4/6 in G1 phase was sufficient to impair origin licensing Supplementary Fig. 11e). When CDK4/6 was inhibited also in G2/M, the origin licensing defect was enhanced, suggesting that licensing efficacy correlates to treatment duration and that a full block

of MCM loading likely requires sustained CDK4/6 inhibition during mitotic exit and G1 phase.

To study how transient loss of CDK4/6 activity affects DNA replication initiation genome-wide, we made use of our time-resolved EdU-sequencing setup in which we allow RPE Cdc6d Cdt1d cells to progress through G1 phase in a synchronised fashion, block CDK4/6 activity either in early or late G1 phase and detect DNA replication in early S-phase cells (Fig. 5a). Flow cytometry analysis indicated that palbociclib treatment during early or late G1 phase is sufficient to cause a significant reduction of EdU-positive cells, confirming the need for sustained CDK4/6 activity to promote DNA replication (Fig. 5b). To define the interplay between CDK4/6 activity and the timing of origin licensing, we combined palbociclib treatments in early or late G1 phase with concurrent or alternating treatments with degron drugs (ASV, DOX, and NAA) and subsequently measured origin activity in early S phase (Supplementary Fig. 11). To directly compare the consequences of the different treatments, we determined the average EdU intensities of thousand top-ranked early S-phase origins for each condition (Fig. 5c). Focusing first on the palbociclib-only data, we found that

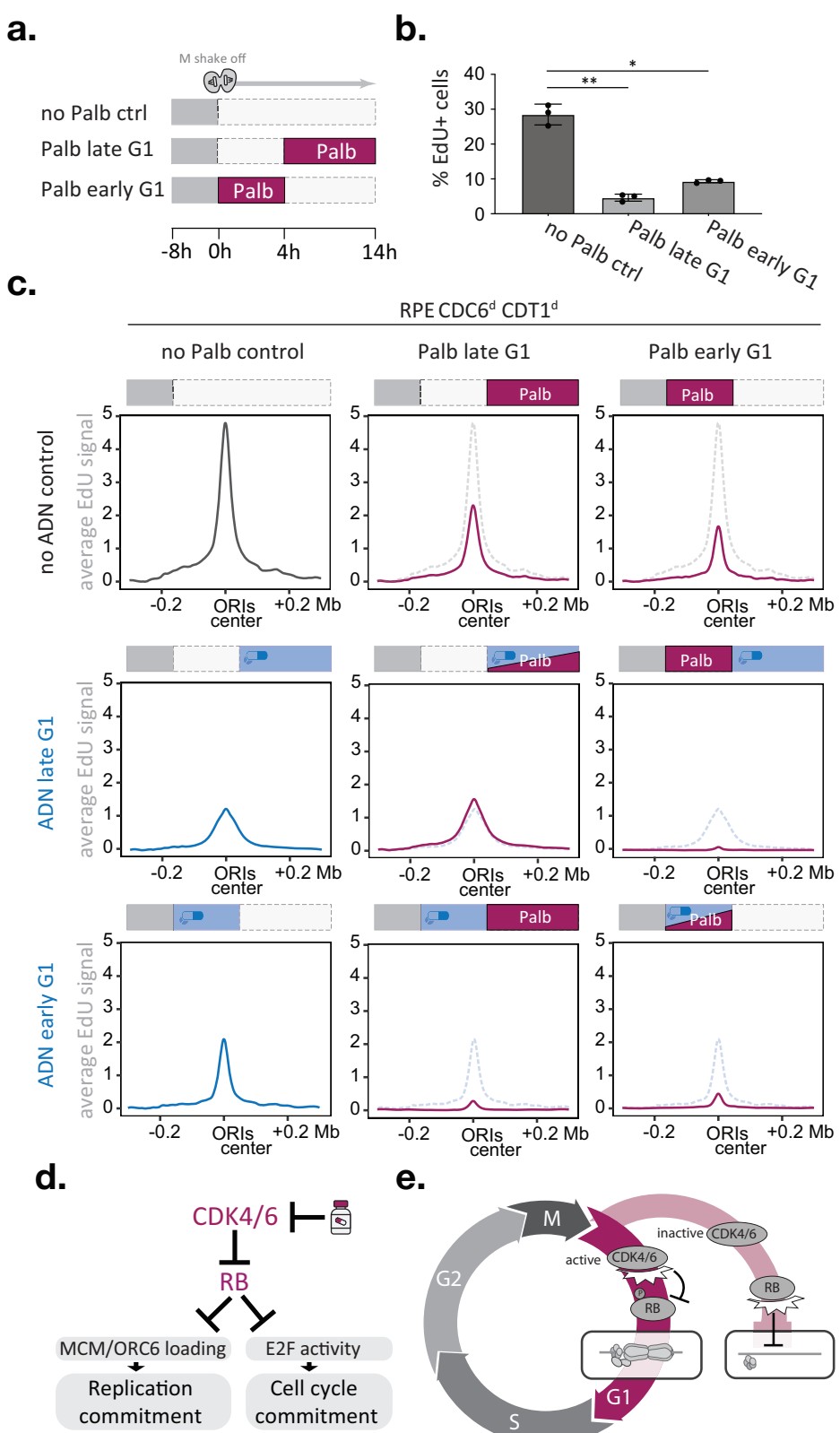

CDK4/6 inhibition during early or late G1 phase diminished average origin activity by 65% or 50%, respectively. Notably, a 4-h palbociclib treatment during early G1 phase decreases origin activity more effectively than a 10-h treatment during late G1 phase, even when the former treatment allows more cells to enter S phase (Fig. 5b, c). Next, we examined how CDC6 and CDT1 degradation affected origin activity. In line with our previous observations, we find CDC6 and CDT1 to be

needed in early and late G1 phase for efficient replication initiation (Fig. 5c, blue lines). We noted that CDC6 and CDT1 degradation until S phase caused average EdU peaks to be lower and wider than controls (Fig. 5c), which matches CDT1's dual role in promoting origin licensing and limiting fork speed in early S phase, respectively[60,61].

Finally, we studied the effect of combining CDK4/6 and CDC6/CDT1 deficiency in a single G1 phase. Simultaneous palbociclib

**Fig. 5 | CDK4/6 activity is needed at the time of origin licensing. a–c** Time-resolved EdUseq upon CDK4/6 inhibition and/or CDC6/CDT1 degradation in early or late G1 phase. **a** Outline of the palbociclib treatment timings. **b** bar graph depicts the percentage of EdU-positive cells by FACS analysis after the treatments indicated in (**a**); error bars indicate SD; black dots indicate replicate means. Statistical significance was determined using a two-tailed paired t test; $n = 3$) and asterisks indicate $p < 0.05$ (* $p = 0.00038$; ** $p = 0.00019$). **c** Average origin activity (i.e., mean $\sigma$ values) at 1 Kb resolution around 1000 predefined, most active early S-phase origins after indicated palbocilib and/or ADN treatments. Schematic experimental outlines are depicted above each graph. Dotted lines indicate the average origin activity of the respective 'no palbociclib' control. **d, e** Proposed working models, **d** FDA-approved CDK4/6 inhibitors impede origin licensing and cell cycle commitment via RB regulation. **e** The CDK4/6-RB axis coordinates origin licensing and cell cycle commitment. In cells with high CDK4/6 activity, pocket proteins such as RB are inhibited by phosphorylation, which promotes both origin licensing and cell cycle commitment, thus ensuring efficient genome replication. In cells with low CDK4/6 activity, pocket proteins such as RB remain active, which stalls origin licensing and cell cycle commitment, thus preventing energy-consuming licensing activities in cells exiting the cell cycle. Source data are provided as a Source Data file.

treatment and CDC6/CDT1 degradation during late G1 phase did not alter origin activity compared to CDC6/CDT1 degradation alone, suggesting that CDK4/6 activity acts in the same pathway as CDC6 and CDT1. Indeed, when CDC6/CDT1 degradation and CDK4/6 inhibition were implemented consecutively, i.e. switching from CDC6/CDT1 degradation during early G1 phase to CDK4/6 inhibition during late G1 phase, origin activity was diminished to 18% of mock-treated controls, which is two to threefold lower than either single treatment (Fig. 5c). The reciprocal experiment, switching from CDK4/6 inhibition during early G1 phase to CDC6/CDT1 degradation during late G1 phase, mirrors these results and reduced average origin activity to 17% of mock-treated controls. These findings demonstrate that efficient origin licensing requires CDK4/6 activity and the MCM loading machinery to be present at the same time. Blocking either of these activities during G1 phase is sufficient to reduce origin activity in early S phase. Protein degron approaches are ideally suited to study immediate effects upon protein loss, but for studying phenotype recovery these approaches rely on the target's expression context. In line with CDK4/6 promoting CDT1 and CDC6 mRNA expression[62,63], adding palbociclib during CDC6/CDT1 degradation in early G1 phase stalled the recovery of origin activity compared to CDC6/CDT1 degradation alone (Fig. 5c). Importantly, our biochemical data showed that palbociclib imposes licensing defects that are not restored by increased CDC6 and CDT1 levels (Supplementary Fig. 4d and Supplementary Fig. 7e) and involve ORC6 loading defects that manifest independently of CDC6 expression control (Fig. 3d and Supplementary Fig. 5c). The notion that short (4 h) palbociclib treatments in early G1 phase effectively diminished origin activity (Fig. 5c) supports a direct and upstream function for the CDK4/6 kinases in origin licensing.

## Discussion

The ability to selectively control origin licensing without causing DNA damage or cell cycle arrest allowed us to separate CDK4/6's role in DNA replication and cell cycle control and effectively trigger mitosis with unreplicated genomes in p53-deficient cells. While here we used this ability to identify the CDK4/6-RB axis as a regulator of origin licensing (Fig. 5d), we believe that blocking DNA replication at its root will be valuable beyond the replication field and allow researchers to address fundamental questions in cell cycle biology, development, DNA repair and 3D chromosome organisation[22,64–67].

While our conditional protein degradation studies demonstrate that human CDC6 and CDT1 are critical for MCM loading and DNA replication initiation, in line with recent in vitro work[68], our CDC6 and CDT1 overexpression studies in Palbociclib-treated cells reveal additional layers of origin licensing control in situ. We conclude that human CDC6 and CDT1 are required but not sufficient of the proper origin licensing in human cells. A recent study targeting MCMBP revealed that impaired MCM complex assembly causes MCM destabilization and selective DNA damage in p53-deficient cells[69]. Our finding that clinically-approved CDK4/6 inhibitors can be used to impair the chromatin-binding of productive origin licensing complexes (Figs. 3, 4) opens up prospects for translating pre-clinical p53-selective responses to real-life therapy settings[55,69,70] and provides an explanation for why long-term CDK4/6 inhibitor treatments result in

selective destabilization of MCM proteins[36]. The implication of RB/p107/p130 as key negative regulators of origin licensing in human cells (Fig. 3) also invites continued biochemical and structural investigations, especially now, as in vitro reconstitution of the origin licensing is entering the scientific domain[68,71].

Our genetic analysis in transformed and untransformed cells demonstrate that RB phosphorylation tightly correlates with MCM loading proficiency and that RB is the critical CDK4/6 substrate to modulate origin licensing dynamics in G1 phase (Fig. 3 and Supplementary Fig. 5). While CDK4/6 initiate site-specific RB phosphorylation to prime replication and cell cycle commitment, CDK1/2 activities complete RB inactivation in late G1 phase through cooperative hyperphosphorylation[12,72]. When CDK4/6 are absent, the cell cycle can still proceed because CDK1/2 can compensate for the loss of CDK4/6 activity[73]. The partial redundancy among mammalian CDKs also explains why CDK4/6 double knockout and even CDK2/3/4/6 quadruple knockout mice can undergo DNA replication and organogenesis, albeit with inefficient RB inactivation, retarded DNA replication initiation and severe developmental defects[73,74]. Due to these redundancies, targeting CDK4/6 by catalytic inhibition or protein loss will have significantly different outcomes. Indeed, we find Palbociclib to be more potent than CDK4/6 PROTACS in reducing RB phosphorylation and origin licensing (Supplementary Fig. 6). We propose that, in the absence of CDK4/6, redundant CDK1/2 activities can release the RB blockade but such activities will at the same time create unfavourable conditions for origin licensing (e.g., by impairing ORC and CDC6 function)[75,76]. These contradicting roles of CDK1/2 provide a raison d'être of a dedicated CDK4/6 family to counteract RB pocket proteins and promote timely origin licensing.

Our time-resolved single-cell and origin activity analysis indicates that licensing occurs throughout the G1 phase and without any temporary predefined sub-G1 licensing period (Fig. 2), supporting models of cumulative origin licensing and stochastic origin activation in human cells[58,77]. As our study concentrates on early S phase origins and control of MCM loading steps, future innovations in single-cell sequencing, origin detection throughout S phase and spatiotemporal control of licensing factors will further improve the detection of human origins and additional levels of licensing regulation[78–80]. While our data indicate that CDK4/6 activity controls licensing proficiency, others have found inverse signals, where licensing proficiency controls CDK4/6 or CDK2 activity[81,82], implying that our observations are part of a fundamental feedback mechanism linking cell cycle commitment to origin licensing. While the origin licensing checkpoint requires p53 to signal to CDK activities[81,82], we find that CDK4/6 activities control origin licensing independent of p53 status (Fig. 3b, c and Supplementary Fig. 4b). Our time-resolved genetics and molecular data indicate that CDK4/6 activity is needed throughout G1 phase and that RB proteins inhibit origin licensing through E2F-dependent and E2F-indepenent functions (Supplementary Figs. 5, 7 and 8). Future studies are needed to map the intricate structural interactions between human pocket proteins and origin licensing complexes and define the exact molecular mechanisms behind RB-mediated licensing inhibition. We propose that the CDK4/6-RB axis has evolved to ensure complete genome replication in cells of rapidly dividing tissues and prevent unnecessary

energy investments in genome-wide licensing in senescent or differentiated tissues (Fig. 5e)[3,83,84].

The finding that CDK4/6 activity promotes origin licensing also has ramifications for cancer medicine. The distinctive success of CDK4/6 inhibitors in a growing number of malignancies implies that the clinically relevant function of CDK4/6 is different from mitotic CDKs, yet most efforts to improve CDK4/6-based therapies are still aimed at its canonical function in cell cycle control[16–18]. Enhancing the cytostatic activity of CDK4/6 inhibitors will provide short-term benefits, yet such treatment remains prone to relapse, drug resistance and immunomodulation[16–18]. The implication of CDK4/6 in replication licensing (this study) and the notion that CDK4/6 inhibitors selectively target cancer subtypes[36,85] provides opportunities to generate more durable responses based on selective cytotoxic cell fates. While RB-deficient cells can avoid the cytostatic activity of CDK4/6 inhibitors, origin licensing remains suboptimal (Fig. 3), providing a cogent explanation to why treatments combining CDK4/6 inhibition and DNA replication stress still synergise in RB-deficient cancers[86,87].

# Methods

## Cell culture and drug treatments

Human hTERT-RPE1 (hereafter referred to as RPE) were obtained from the American Type Culture Collection (ATCC; CRL-4000) and cultured in an ambient-controlled incubator at 37 °C and 5% $CO_2$ and maintained using DMEM-F12 GlutaMAX (ThermoScientific, 31331093) supplemented with 1% Pen/Strep (ThermoScientific, 15140122) and 10% heat-inactivated foetal bovine serum (Sigma, F7524). The p53 deficient and isogeneic parental RPE cells are described previously[88] and were kindly provided by Dr. Arne Lindqvist (KI). The RB1-deficient and isogeneic parental RPE cells carrying a nuclear mTurquoise2 reporter are described previously[40] and were kindly provided by Dr. Titia de Lange (The Rockefeller University). The RB1/RBL1/RBL2 triple knockout RPE line[41] was a kind gift from Dr. Hein te Riele (NKI). The non-transformed BJ cells were obtained from the American Type Culture Collection (ATCC; CRL-2522) and maintained at low passage using DMEM Gluta-MAX (Invitrogen) supplemented with 1% Pen/Strep (ThermoScientific, 15140122) and 10% heat-inactivated foetal bovine serum (Sigma, F7524). HBEC CDC6 Tet-ON cells were maintained as previously described[45]. All cell lines were regularly tested for mycoplasma using the LONZA MycoAlert detection kit. The following small-molecule drugs were used at the indicated final concentrations, unless specified otherwise: mimosine (Sigma; 400 µM), palbociclib (200 nM), Trilaciclib (Biosynth; 1 µM), doxycycline (Sigma; 500 ng/ml), 1-Naphthaleneacetic acid (Sigma; 100 µM), Asunaprevir, (Sigma; 1 µM), hydroxyurea (Sigma; 2 mM), MG132 (Sigma; 10 µM), RL5a (Sigma, 2 µM). Mock-treated controls were exposed to equimolar media/solvent concentrations.

## Cloning and plasmid transfections

The pRosa_TeT_OsTIR1_ARF16_Bleo plasmid was generated by cloning a custom-designed OsTIR_T2A_OLLAS_NLS_ARF16-PB1 construct into our previously verified TIR knock-in vector pROSA26-DV1_OsTIR[22] using EcoRI and AgeI sites. To target the human Rosa26 safe harbour locus, we cloned PX458_ROSA by inserting gRNA 5′ GACCTGCTA-CAGGCACTCGT 3′ into PX458_Cas9_GFP (Addgene 48138) using BbSI sites. For tagging endogenous CDC6, we used a verified hCDC6_mAID_SMASh_T2A_Neo plasmid[22] and PX458_Cas9_GFP (Addgene 48138) in which we inserted gRNA GCCAGCTGAA-TACTTTCGGG using BbSI sites. For tagging endogenous CDT1 we generated a hCDT1_mAID_SMASh_T2A_Neo plasmid by replacing the left and right homology arms of hCDC6_mAID_SMASh_T2A_Neo plasmid[22] with custom-synthesised 700–800 bp Cdt1 homology sequences using PciI/SacI and SbfI/SalI sites, respectively. Two independent Cdt1 targeting plasmids were made by inserting gRNA 5′ GTCTGTCCACAGTGGCCCCC 3′ or 5′ GGGGCCACTGTGGACAGACG 3′

into PX458_Cas9_GFP (Addgene 48138) using BbSI sites. To allow subsequent reciprocal tagging of CDC6 and CDT1, we generated hCDC6_mAID_SMASh_T2A_BSD and hCDT1_mAID_SMASh_T2A_BSD plasmids by exchanging the Neomycin resistance cassettes of hCDC6_mAID_SMASh_T2A_Neo and hCDT1_mAID_SMASh_T2A_Neo plasmids, respectively, with custom-synthesised Blastycidin resistance constructs using AvrII/SalI sites. The MCM7-CT plasmid was generated by cloning a custom-synthesised MCM7-CT sequence[20] and a P2A linker into a GFP expression vector (Addgene #17653) using BamHI and EcoRI sites. Constructs are verified by whole plasmid sequencing and are available upon request.

## Generation of RPE CDC6$^d$ CDT1$^d$ lines

We generated two independent RPE CDC6d CDT1d lines using targeted CRISPR/Cas-9 genome editing. We first integrated a DOX-inducible OsTIR and ARF16-PB1 co-expression construct at the human Rosa26 locus (chromosome 3) and subsequently tagged both alleles of CDC6 and CDT1 by an mAID_SMASh double degron. RPE p53-/- cells[88] were transfected with PX458_ROSA and pRosa_TeT_OsTIR1_ARF16_-Bleo using *FuGENE 6 (Promega)* and three days later subjection to Zeomycin (Invitrogen) selection. Single colonies were isolated using Pyrex Cloning Cylinders (Merck) and successful knock-ins were verified by PCR using QuickExtract™ DNA Extraction Solution (Lucigen) and primers 5′ ACCTCAGATCCAATTCTCTG 3′ (Rosa locus) and 5′ GGTCGGAGGTCGTGTCCACG 3′ (Bleo insert). Doxycycline-dependent ARF16-PB1 and OsTIR expression was verified by IF and Western blotting using anti-OLLAS (NBP1-06713; Novus) and anti-Myc-Tag antibodies (#2276; Cell Signaling). RPE OsTIR ARF16-PB1 line #11F was transfected using XtremeGene9 (Sigma-Aldrich), donor plasmids hCDC6_mAID_SMASh_T2A_Neo or hCDT1_mAID_SMASh_T2A_Neo and PX458_Cas9_GFP-derivative plasmids expressing Cdc6 or Cdt1 gRNAs, respectively. Three days post-transfection GFP+ cells were sorted and subjected to Geneticin selection (G418; Gibco). Clonal cell lines were established by limiting dilution in 96-well plates and successful degron-tagging was verified by PCR, IF and degron-induced growth arrest. For the RPE CDC6d CDT1d clone # III/A1-1, hereafter clone #1, CDC6 was tagged first and then CDT1, while for the RPE CDC6d CDT1d clone # II/A2-3, hereafter clone #2, CDT1 was tagged first and then CDC6. To tag the second gene, the cells were transfected using Xtre-meGene9 (Sigma-Aldrich), donor plasmids hCDC6_mAID_SMASh_-T2A_BSD or hCDT1_mAID_SMASh_T2A_BSD and PX458_Cas9_GFP-derivative plasmids expressing Cdc6 or Cdt1 gRNAs, respectively. Three days post-transfection GFP+ cells were sorted and subjected to Blasticidin S (Thermo Fisher) selection. Single colonies were isolated using Pyrex Cloning Cylinders (Merck) and successful knock-ins were verified by Western blot and IF.

## Immunofluorescence assays and microscopy

For QIBC analysis, RPE cells were seeded in 96-well imaging plates (Sigma) and 10 mM EdU (5-ethynyl-2′-deoxyuridine, Jena Bioscience) was added to live cells for 1 h prior to fixation. After treatment, cells were washed in TBS supplemented with 0.1% Tween20 (hereafter referred to as TBS/T) and DPBS (#2037539 GIBCO), fixed in 4% Formaldehyde solution (#02176; Histolab) for 7 min, permeabilised in cold methanol (Sigma-Aldrich) for 2 min, washed in TBS/T and DPBS and incubated in blocking media (TBS/T and 2% bovine serum albumin) for 1 h. To detect chromatin-bound proteins, cells were pre-extracted for 90 s using ice-cold extraction solution (0.5% Triton-X-100 in CSK buffer) before fixation. Fixed samples were incubated with the primary antibody (Cyclin A2 1:400 #66391-1-Ig; Proteintech, MCM2 1:400; #4007; CST or MCM6 1:400 sc-393618; SantaCruz) in blocking media overnight at 4 °C, washed in TBS/T and DPBS and incubated with the secondary antibody (Alexa Fluor 555 anti-Mouse 1:800 #A21422, Life Technologies or Alexa Fluor 488 anti-Rabbit 1:800 #A11008, Life Technologies) and 50 ng/ml DAPI (#D1306;

ThermoFisher Scientific) for 1 h at room temperature. Samples were washed in TBS/T and DPBS and EdU-Click chemistry was performed by incubation in 100 mM Tris, 1 mM CuSO4 (C1297; Sigma), 100 mM ascorbic acid (#A4544 Sigma) and fluorescent dye azide (#A10277, Invitrogen) for 1 h at room temperature, then washed in TBS/T and DPBS. Stained samples were stored in DPBS. Images were acquired at room temperature using Nikon Ti2 ECLIPSE microscope (20X air objective) and analysed using custom CellProfiler and R pipelines. To quantify the fraction of anaphases, cells were treated as indicated, fixed and stained as above using anti-Pericentrin antibody (1:400; #ab4448; Abcam), anti-Rabbit antibody (1:800, #A11008, Life Technologies) and DAPI. More than 50 DAPI-stained mitotic nuclei were segmented for each condition (in quadruplicate) and classified as anaphase (showing two separate DAPI masses) or pro/metaphase (showing a single DAPI mass). Confocal data were acquired on a Zeiss LSM 780, equipped with a Plan-Apochromat 63×/1.4 oil immersion objective (Carl Zeiss). Optical sections were acquired with 43 nm × 43 nm pixel size and 130 nm step size. Images were deconvolved with Huygens software (Scientific Volume Imaging).

## Immunoblotting

For Western blot analysis of whole lysates, RPE cells were seeded in 6-well or 10 cm culturing plates (Sarstedt), treated as indicated, washed with DPBS and lysed in RIPA buffer supplemented with protease and phosphatase inhibitors (Thermo Fisher Scientific, 78444) and sonicated for five cycles of 30 s on and 15 s off, in a Bioruptor® (Diogenode). Protein concentrations were quantified using a DC™ Protein Assay Kit II (Bio-Rad, 5000112) and lysates were boiled in Laemmli sample buffer for 5 min at 95 °C before loading onto SDS-PAGE gels. Proteins were transferred onto nitrocellulose membranes using the Trans-Blot SD Semi-Dry Transfer System (BioRad, 170-3940), blocked in TBS/T containing 2% skim milk and probed using primary antibodies and HRP-conjugated secondary antibodies. Chemiluminescence signals were detected using SuperSignal™ West Dura (Thermo Fisher Scientific, 34076) and an Amersham Imager 600 scanner. The Pierce Subcellular Protein Fractionation Kit (78840) was used for the extraction of chromatin-bound, nuclear soluble and cytoplasmic fractions, following the manufacturer's instructions. Immunoblot analysis was performed a minimum of two times with independent biological replicates and ImageJ software was used for protein signal quantification.

## Antibodies

All antibodies used in this study are described in Supplementary Table 2.

## Flow cytometry

To identify the efficiency of S phase entry after induction of Cdc6 and Cdt1 degradation, the asynchronous parental RPE or RPE CDC6d CDT16d cells were initially incubated in the presence of 1 μg/ml doxycycline-hydrochloride (Sigma, Cat. No. D3447) for 4 h and then 250 μM NAA and 200 nM ASV were added to the culture medium for 2 days. Before being collected and fixed overnight in 90% methanol, cells were incubated in 25 μM EdU (Invitrogen, Cat. No. A10044) for 30 min. Following, cells were perrmeabilized for 30 min in DPBS supplemented with 0.2% Triton X-100, stained for EdU content using the Click-iT Kit (Invitrogen, Cat. No. C-10424) for 30 min at RT and stained for the DNA content using the Propidium Iodine (Sigma, Cat. No. P4170) while treated with RNAse (Roche, Cat. No. 11119915001) overnight at RT. Finally, flow cytometry was performed (Gallios, Beckman Coulter) and EdU-DNA content profiles were acquired. To measure the percentage of EdU-positive cells at specific times after mitotic exit, RPE CDC6d CDT16d cells were initially synchronised in mitosis by exposure to 200 ng/ml nocodazole (Tocris, Cat. No. 1228) for 8 h. The mitotic cell population was isolated by mitotic shake-off and released into the cell cycle for the indicated time period in the presence of 25 μM EdU (Invitrogen, Cat. No. A10044). When Cdc6 and Cdt1 degradation was induced, 250 μM NAA and 200 nM ASV were added to the culture medium for the desired time window. 4 h prior to the degron drugs addition, cells were treated with 1 μg/ml doxycycline-hydrochloride (Sigma, Cat. No. D3447). For Cdk4/6 inhibition, 150 nM palbociclib was added to the culture medium for the indicated timeframe. After the desired release period, cells were harvested and fixed overnight in 90% methanol. The cells were then permeabilized in DPBS supplemented with 0.2% Triton X-100 for 30 min and stained for EdU content for 30 min at RT using the Click-iT Kit (Invitrogen Cat. No. C-10424). Genomic DNA was stained with Propidium iodine (Sigma, Cat. No. P4170) in combination with RNAse treatment (Roche, Cat. No. 11119915001) overnight. Lastly, cells were subjected to flow cytometry and EdU-DNA content profiles were acquired. To acquire the kinetics for M phase entry after Cdc6 and Cdt1 degradation, cells were initially synchronised in mitosis by exposure to 200 ng/ml nocodazole (Tocris, Cat. No. 1228) for 8 h. The mitotic cell population was isolated by mitotic shake-off and released for 16 h in the presence of 2 mM thymidine (Sigma, Cat. No. T1895). Then, thymidine was removed, and cells were released in the cell cycle for 1 h or 10 h. Cdc6 and Cdt1 degradation was induced by addition of 250 μM NAA and 200 nM ASV from the moment of mitotic exit until cells were harvested, with a preceding incubation of cells in 1 μg/ml doxycycline-hydrochloride (Sigma, Cat. No. D3447) for 4 h. After collection, cells were fixed overnight in 90% methanol, permeabilized in DPBS supplemented with 0.2% Triton X-100 for 30 min at RT and blocked in 1% BSA-PBS for 1 h at RT. Following, cells were incubated in primary antibody against phospho-histone H3 (Ser10) (H3S10p) (Cell Signalling, Cat. No. D7N8E, 1:1600) for 1 h at RT and the secondary Goat anti-Rabbit IgG, Alexa Fluor™ 647 antibody (Invitrogen, Cat. No. A-21244, 1:200) for 30 min at RT. The genomic DNA was stained by Propidium iodine (Sigma, Cat. No. P4170), and cells were treated with RNAse (Roche, Cat. No. 11119915001) overnight, with subsequent flow cytometry analysis.

## EdUseq

To obtain the replication initiation profiles of RPE III/A1-1 cells after Cdc6 and Cdt1 degradation during specific time periods of G1 phase, the EdUseq assay was followed, as described previously[89]. Briefly, cells were synchronised in mitosis by 200 ng/ml nocodazole (Tocris, Cat. No. 1228) for 8 h, the mitotic cell population was isolated by mitotic shake-off and released for the indicated time period in presence of 25 μM EdU (Invitrogen, Cat. No. A10044) and 2 mM Hydroxyurea (HU) (Sigma, Cat. No. H8627). For Cdc6 and Cdt1 degradation, cells were treated with 250 μM NAA and 200 nM ASV for the desired timeframe. 4 h prior to the degron drugs addition, cells were incubated with 1 μg/ml doxycycline-hydrochloride (Sigma, Cat. No. D3447). When Cdk4/6 was inhibited, 150 nM palbociclib were added to the culture medium for the indicated time frame. After harvested cells were fixed in 90% methanol overnight, permeabilized in DPBS supplemented with 0.2% Triton X-100 and the EdU biotinylation was performed using the biotin-azide linker (Azide-PEG(3 + 3)-S-S-biotin) (Jena Biosciences, Cat. No. CLK-A2112-10) and the Click-iT Kit (Invitrogen, Cat. No. C-10424). Following, gDNA extraction by phenol/chloroform, ethanol precipitation and sonication were performed. The EdU-labelled DNA was isolated by Dynabeads MyOne streptavidin C1 (Invitrogen, Cat. No. 65001) and used for library preparation using the TruSeq ChIP Sample Prep Kit (Illumina, Cat. No. IP-202-1012). One hundred base pair single-end read sequencing reactions were run on an Illumina Hi-Seq 2500 or Illumina Hi-Seq 4000 sequencers. The obtained sequence reads were aligned to human genome assembly (GRCh37/hg19) using the Burrows–Wheeler aligner algorithm and were further analysed by custom scripts.

**Reporting summary**

Further information on research design is available in the Nature Portfolio Reporting Summary linked to this article.

## Data availability

The fastq sequencing data and associated information described in this study have been deposited to NCBI's Gene Expression Omnibus with the GEO series accession number GSE278714 [https://www.ncbi.nlm.nih.gov/geo/query/acc.cgi?acc=GSE278714] (see Supplementary table 1 for file descriptions). All remaining data generated or analysed during this study are included in this published article and its supplementary information files. Material requests should be addressed to the corresponding authors. Source data are provided with this paper.

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

## Acknowledgements

We acknowledge Ana Agostinho at the Advanced Light Microscopy facility at KTH-SciLifeLab, part of the National Microscopy Infrastructure, NMI (VR-RFI 2019-00217), for imaging support. The authors thank Gabor Merényi, Yiqiu Yang, Sophie Boersma, Srinidhi Rengarajan, Matilde Fonseca and Demetrio Turati for technical assistance and the Genome Biology division for valuable discussions and technical support. This work was supported by grants from the Swedish Research Council to B.L. and J.B., Swedish Cancer Society to B.L. and J.B., the European Commission (REPLISTRESS) to T.D.H., the Swiss National Science Foundation (grant nos. 182487 and 186230) to T.D.H., Karolinska Institutet to B.L. and The Mark Foundation for Cancer Research (ASPIRE Spr21-25 and ASP-II-0629263208) to B.L. and J.B.

## Author contributions

B.L. supervised the project and conceived the study. B.L. and T.D.H. designed the experiments. A.S.P., M.H., A.N., A.H.S., P.B., G.G. and B.U. performed the cell-based experiments. B.L. designed and generated the described double-degron cell models. M.H., A.H.S., P.B., G.G. and B.U. performed the single-cell imaging studies. A.N. and A.H.S. performed the biochemical assays. A.S.P. and S.N. performed the FACS studies and processed samples for sequencing. T.D.H., M.H., B.U., and V.S.D. performed the bioinformatic analyses. A.S.P., M.H., J.B., T.H.D. and B.L. analysed the data, and all authors commented on the manuscript.

## Funding

## Competing interests

T.D.H. is the founder and stockholder of FoRx Therapeutics. The remaining authors declare no competing interests.
