## [Transparent Peer Review file · Nature Communications]

Temporal control of human DNA replication licensing by CDK4/6-RB signalling and chemical genetics

Corresponding Author: Dr Bennie Lemmens

Version 0:

Reviewer comments:

Reviewer #1

(Remarks to the Author)

This manuscript investigates the role of CDK4/6 activity in licensing origins in G1 phase. The main concern is the overall premise of the paper since it is well established from gene knockout studies (CDK4/6 and cyclin Ds), as well as from many previous CDK4/6 inhibitor studies, that mammalian cells can license origins and replicate DNA if they lack CDK4/6 activity or have CDK4/6 inhibited. In most human non-transformed and cancer cells, CDK4/6 inhibition generally delays but does not fully prevent origin licensing and DNA replication and proliferation.

As an added comment, it is well established that E2F must induce both CDC6 and CDT1 to trigger licensing, and they do not show whether this dual induction is sufficient to trigger licensing (they only induce CDC6). Moreover, they did not identify what the proposed CDK4/6 target is that could explain a potential role of CDK4/6 (in addition to its role of regulating Rb). However, the role of CDK4/6-regulated Rb phosphorylation in activating E2F is well established and could explain the lower licensing. Notably, their own, as well as many other people's Rb null data, shows that E2F activation is sufficient to license origins without a need for CDK4/6 activity. Also, Cyclin E-CDK2 activation and loss of Rb are main CDK4/6 relapse mechanisms in various cancers; both these pathways increase E2F activity when CDK4/6 is inhibited – which is consistent with E2F activation and not CDK4/6 activity directly being responsible for inducing CDC6 and CDT1 and triggering origin licensing.

1. Logical inconsistencies:

a. The authors make the claim that CDK4/6 activity in G1 is required to license origins of replication in G1. However, it is well established that cells can proliferate without any CDK4/6 activity, and origin licensing is required for proper cell cycle progression. Further, their own data in Figures 3e,f show that in the presence of Palbociclib and Mimosine, when the RBs are knocked out, cells can still load MCs onto DNA, again arguing that CDK4/6 activity is not required for origin licensing but E2F is.

b. Since all mammalian licensing factors (especially Cdt1 and CDC6) are highly regulated by E2F, isn't a simpler explanation consistent with the literature and their data that origin licensing is triggered by E2F activation? This likely also requires that APC/CCdh1 is active to prevent geminin accumulation (which blocks Cdt1) but APC/CCdh1 is generally active until G1/S.

c. Since licensing requires E2F activation to make CDC6 and Cdt1, it seems likely that the lack of licensing is the result of delayed origin licensing in CDK4/6 inhibited cells.

In this regard, the timing when they are evaluating origin licensing with Palbociclib treatment is likely not long enough time to allow the CDK4/6-independent pathways to be increasing E2F activity (as previously shown cells with inhibited CDK4/6 are entering the cell cycle more slowly by using gradual CDK2 activation but only after some 24 hours).

d. In Figure S5b, they only overexpress CDC6, but this experiment is not enough to conclude that CDK4/6 activity is needed since CDC6 is only part of the known trigger. This conclusion can only be reached by overexpressing both Cdt1 and CDC6 since both are required for origin licensing. Also, there might potentially be additional E2F-regulated factors needed for licensing (e.g. some MCs are E2F regulated). Again, the existing evidence shows that E2F induction of CDC6 and CDT1, but not CDK4/6 activity itself, are minimally needed as triggers for origin licensing. CDK4/6 is important indirectly, by activating E2F.

2. Novelty:

- a. Overall, the authors do not show sufficient novel mechanisms what CDK4/6 might be doing in G1. The delay in both cell cycle entry and G1 progression with CDK4/6 inhibition has already been extensively described. As they have mentioned, the requirements of CDC6 and Cdt1 in origin licensing have also been extensively characterized.
- b. If CDK4/6 indeed plays a direct role in licensing, they do not show which sites on which of the regulators of origin licensing has to be phosphorylated (CDK4/6 does not have many targets beyond Rb). More likely is that they are using in their CDK4/6i experiments conditions where E2F activity is still low or delayed in G1, explaining why they do not see origin licensing in their CDK4/6 inhibited conditions since E2F activation is delayed and not due to a new regulatory mechanism.

3. Other points:

- a. The experiments where they overexpress C-terminal MCM7 shows reduced total RB levels, which may account for the reduction in phosphorylated RB, and the reduction in MCM2 signal (Figure S6d). The reason for this difference in total RB has not been further investigated and may be relevant. For example, the decrease in total RB may cause a reduction in detectable phospho-RB. To exclude that this is the explanation, they could use the ratio phospho-Rb over total Rb.
- b. The results also are shown in RPE-1 cells, and need to be shown in other cell lines. Specifically, they should use of cancer or other cell lines that lack CDK4/6, lack RB, or are insensitive to Palbociclib treatment - this could be mechanistically informative and may change the interpretation of their data.

Reviewer #2

(Remarks to the Author)

Piscitello et al. investigated the effect of CDK4/6-Cyclin D inhibitors on the replication licensing (i.e. MCM-loading onto chromatin) in G1 phase. First, the authors established an RPE1 degron cell line in which CDC6 and CDT1 could be conditionally degraded, demonstrating the MCM-loading was strongly inhibited upon the degradation of CDC6 and CDT1 (Fig. 1). Using this degron cell line, they showed the MCM-loading occurred throughout the G1 phase with no preference for specific origins to be licensed in specific G1 periods (Fig. 2). Next, the authors examined the effect of the CDK4/6 inhibitor palbociclib, revealing that it reduced the MCM-loading in G1 phase (Fig. 3a,b,c). Importantly, this licensing inhibition was rescued in cells lacking three Rb family proteins (Fig. 3d,e,f), indicating that the RB-family proteins mediate licensing inhibition caused by palbociclib. By combining palbociclib treatment with CDC6/CDT1 degradation, they demonstrated that both CDK4/6 activity and CDC6/CDT1 are simultaneously required for efficient licensing and DNA replication (Fig. 4, 5). Finally, the authors concluded that the CDK4/6-Rb axis is critical for MCM-loading in G1, thus for efficient DNA replication in S phase.

The study is carefully executed and presents high-quality data. While some results, such as mitotic cells without DNA replication (Fig. 1e and S1c) and the exploration for specific replication origins licensed at differential G1 time periods (Fig. 2 and S3), may not directly support the central conclusion, they offer valuable insights to the replication and cell-cycle research fields.

The key finding is that the CDK4/6-Cyclin D/Rb pathway promotes the MCM-loading in G1 phase. CDK4/6-Cyclin D is known to phosphorylate Rb in G1 phase, releasing the E2F transcription activator from Rb and inducing the transcription of licensing factors, such as CDC6, CDT1 and MCM5/6 (Yan et al. PNAS, 1998; Ohtani et al. Oncogene, 1998; Hatenoer et al. MCB, 1998; Yoshida and Inoue, Oncogene, 2004; Ohtani et al. Oncogene, 1999).

However, it remains unclear to me whether MCM-loading is mediated solely through E2F-dependent transcription or if CDK4/6-Cyclin D/Rb has a direct role in this process (Fig. 5d,e). The data supporting a direct role are Fig. 3c and Fig. S5, which show the chromatin-bound MCM2 levels were reduced despite the total nuclear MCM levels being maintained after palbociclib treatment for 18 h (Fig. 3c) and palbociclib treatment reduced MCM-loading in cells CDC6 was overproduced (Fig. S5).

To strengthen the conclusion that the CDK4/6-Rb pathway has a direct role in promoting MCM-loading in G1 phase, the authors should assess the total and chromatin-bound levels of CDC6, CDT1 and MCMs by Western blotting and flow cytometry under the conditions presented in Fig. 3a, d, e, and f. This analysis would help distinguish between transcriptional and direct regulatory roles of the CDK4/6-Cyclin D/Rb axis in licensing. Revise the model presented in Fig. 5d,e.

Minor points

- In Fig. S7b, the authors showed that p53 positive cells did not enter mitosis when licensing was inhibited by palbociclib and ORC inhibitor RL5a. I did not see any explanation related to this result. Can you discuss why p53 positive and negative cells showed differential phenotypes?

- In Fig. 3d,e,f, it is not clear how long the cells were treated with mimosine and palbociclib. The experimental condition should be indicated in the figure legend. These are key data in this paper.

- The authors should refer to the supplementary figure panels in the main text rather than just referring to Fig. S1 or S2. It is difficult to look at the supplementary figure while I read the main text.

Reviewer #3

(Remarks to the Author)

Review of Sesenko Piscitello et al.

This is a very interesting study on important and inter-related unresolved questions about the mammalian cell cycle: whether origin licensing occurs at a specific point in G1 phase, and whether it depends on CDK4/6 activity. The authors use an innovative combination of degron-tagging of two licensing proteins to chemically control replication licensing at different stages post-mitosis and block-release experiments with CDK4/6 inhibitors and/or mimosine, a drug that prevents S-phase onset, followed by analysis of replication using single-cell techniques with genomic resolution, as well as analysis of chromatin loading of replication factors. They reveal some very novel observations, that are potentially of considerable interest: most importantly and surprisingly, origin licensing occurs throughout G1-phase, in contrast to the prevailing dogma that they occur at a specific point in early G1 in a window of low CDK activity; different specific origins do not have a particular timing of licensing in G1; and origin licensing may at least partially depend on CDK4/6-mediated RB phosphorylation, at least in this system. The latter is perhaps the weakest of the conclusions (see below), and so it is a shame that the title of the paper focused on it. Using genetic knockout of paralogues, they further note a partial redundancy of RB family proteins in this control and show evidence of a potential mechanism by which RB family protein phosphorylation may work. All of these are very useful observations that could potentially contribute to establishing a new paradigm about this critical phase of the cell cycle.

I feel, however, that some revisions are required. First, I think the CDK4/6i experiments have been slightly misinterpreted because of the way they were performed. This hinges around the claim made in the title: "Licensing of human DNA replication origins requires CDK4/6 activity in G1 phase". On an aside, by talking uniquely about human cells, this sidesteps the uncomfortable question about why the cell cycle, and DNA replication, is unaffected by genetic knockout of both CDK4 and CDK6 in mice. This work was neither mentioned nor cited, whereas it seems unlikely that mouse cells do anything very differently to human cells, and I think there are ways of reconciling the data of the current paper with the mouse data (see below). Equally importantly, I would suggest that it is unlikely that CDK4 and CDK6 do anything specific in G1, and this also appears to have been overlooked. Work from former members of Tobias Meyer's team (Steven Cappell and Sabrina Spencer) has shown that CDK4 and CDK6 operate throughout the cell cycle; in particular, Sabrina Spencer's team has shown evidence that pre-mitotic CDK4/6 activity affects the following G1. Her original data published as a postdoc in the Meyer lab also do not fit with the traditional but probably outdated model (see reviews by Steven Cooper) of the "restriction point" in G1, since serum withdrawal immediately after anaphase does not prevent cell cycle progression. To come back to the current paper, I think it is very likely that if CDK4/6 had been inhibited specifically prior to mitosis, or at any point after beginning of the previous S-phase, they would also see a defective G1-S phase progression in the following cell cycle due to insufficient origin licensing. Thus, I think it is premature to state that origin licensing requires CDK4/6 activity in G1. That it requires CDK4/6 activity at some point in the cell cycle, however, seems likely, and some experiments would easily address this.

Second, the conclusion of a specific block of licensing by CDK4/6 inhibition is not completely convincing. While in Fig 3a, the reduction in chromatin-bound MCM is obvious, CDC45 loading (which is downstream of MCM) is only slightly reduced, and is not further reduced by a combination of CDK4/6i that give a tighter block over RB phosphorylation. DNA replication was apparently not analysed in this experiment, and so it cannot be concluded that CDK4/6i block licensing. It is also surprising that chromatin loading of the licensing factors CDC6 and CDT1 was not assessed in these experiments. Indeed, the point at which the MCM loading defect occurs is not entirely clear, partly since some experimental conditions used CDK4/6i alone (eg Fig 3a lanes 1-3 and 5) and assessed some factors, while others used CDK4/6i on top of mimosine (eg Fig 3b, d-f, Fig S5b-d) and assessed an additional factor, namely, ORC6, whose chromatin loading is required for CDC6/CDT1 loading. It is striking that ORC6 levels decrease upon CDK4/6i treatment. The experiments performed to address the stage of the block, and shown in Figure 4, were interpreted as showing that CDK4/6i inhibit licensing upstream of CDC6/CDT1 function, since loss of the latter two proteins upon CDK4/6i release did not allow DNA replication, whereas the control could still replicate. However, it is an experiment in asynchronous cells, in which CDK4/6i will act at all stages of the cell cycle, and it is possible that as a consequence some cells have not passed mitosis, while other cells having passed mitosis have not synthesised sufficient factors required for licensing (ORC, CDC6, CDT1). This, and their loading onto chromatin, should be assessed. The complementary experiment has been performed: blocking cells by degrading CDC6/CDT1, and then releasing into palbociclib (Fig 5c), but unfortunately licensing was not analysed in this experiment. If it had been done, it would allow to rule out that CDK4/6i prevent MCM loading to already-bound CDC6/CDT1. DNA replication is still inhibited, as expected, since origin firing and thus DNA replication require the triggering of licensed origins by other CDK (CDK1 or CDK2), and CDK4/6i inhibits the expression of cyclins required to activate them. As it stands, it cannot be concluded from Fig 5 experiments that CDK4/6 activity is required at the time of origin licensing, although these experiments are very interesting since they suggest that CDK4/6 is not required at a particular "restriction point" but acts throughout G1 (if not the entire cell cycle).

How can the interpretation of the authors be reconciled with the observation that CDK4/6 knockout in mice does not affect DNA replication? Likely because upon genetic knockout of CDK4/6, or even CDK2/3/4/6 (Santamaria et al., 2007), CDK1 steps in to rescue RB phosphorylation. CDK4/6 inhibition is qualitatively different in that cyclin D is still present and can still interact with RB, but can no longer phosphorylate it. This should also be discussed, and it might also necessitate a modification of the title: CDK4/6 are not necessarily required for licensing, because in their absence CDK1/2 can do it. Instead, one could say that CDK4/6 promote licensing, or, an alternative explanation that could perhaps fit better with the data in the paper, they promote the acquisition of a licensing-competent state by allowing synthesis of the licensing factors themselves (which would be novel and interesting). This of course requires confirmation as described above.

Notwithstanding these criticisms, I think the findings on origin control and CDK4/6 action throughout G1 (Figures 1, 2, 5) are extremely interesting and important and would deserve a very visible publication. In conclusion, I think this is an excellent and important study but requires some work before publication.

Version 1:

Reviewer comments:

Reviewer #1

(Remarks to the Author)

The authors have addressed my main issues.

Reviewer #2

(Remarks to the Author)

The authors added new data supporting the evidence that CDJ4/6-Cyclin D has a direct role in origin licensing, mediated by the RB family proteins. They have clearly addressed my previous concerns.

To make the final version, the authors should double-check the order of Figures S5 to S7, as the text on pages 7-8 does not align with these figures. Probably, Figure S5 should be Figure S7. Additionally, Figures 4d and S10 are missing scale bars.

Reviewer #3

(Remarks to the Author)

Lemmens, Halazonetis and colleagues have taken into account comments of all reviewers and performed useful additional experiments to address the most important points raised. This is indeed an improved manuscript since it now shows strong evidence that the phenotypes of loss of RB/RBL activity on licensing are partly direct, ie independent of E2F-mediated transcription, which was a major criticism of reviewers 1 and 2. This can of course still be debated but I agree that the balance of evidence points to direct effects that are at least independent of CDC6 and CDT1 levels.

The authors also now address, mostly by discussion, the model whereby CDK4/6 can be compensated for by CDK1/2 in the absence of the former. I would have a different interpretation: namely that CDK4/6 promote CDK1/2 activity (not necessarily just by promoting E2F transcription), and either activity can promote licensing and/or firing, with nothing special about CDK4/6 other than the fact that it is downstream of growth regulating pathways and upstream of CDK1/2. The fact remains that the authors have not formally demonstrated that CDK4/6 do not also do something downstream of CDC6/CDT1 chromatin loading/licensing, and, since they have only tested CDK4/6 inhibition, they have also not formally excluded that CDK1/2 inhibition might have similar effects on licensing as CDK4/6 inhibition (though it will also inhibit firing). But both models are useful and further studies in the field will indicate which is correct.

I would consider changing the title again because it has now become a bit of a non-sequitur, it mixes temporal control by the cellular biochemistry (CDK4/6/RB) and rapid protein degradation, without specifying that the latter is an experimental system and not a cellular control. It also, in my view, fails to fully capture the impact of the study. It would read better something like: "Rapid protein degradation in human cells reveals direct promotion of DNA replication licensing throughout G1 by CDK4/6-RB signalling". Finally, there are quite a few typo or English errors in the manuscript (eg lines 172, 274, 334, 348, 360, 375, 419, 455). Once these are corrected, I do not see any reason to delay the publication of this manuscript, which makes an important contribution to understanding cell cycle control and DNA replication in particular.

Response to the reviewers concerning revised manuscript NCOMMS-24-81972A

We thank the reviewers for their thorough evaluation and valuable feedback. We are pleased that the reviewers recognized the novelty, experimental approach, and broad implications of our study.

The prevailing dogma holds that replication origin licensing is repressed by cyclin-dependent kinase (CDK) activity and occurs at a specific point in early G1, during a window of low CDK activity, to ensure that origins are licensed only once per cell cycle (Basu et al Nature 2022 PMID: 35676478; Amasino et al. 2023 PMID: 37428921). While two major CDK families—CDK1/2 and CDK4/6—have evolved to control cell cycle commitment, their precise roles in regulating origin licensing in human cells remain unclear, particularly regarding the involvement of the retinoblastoma (RB) protein and origin recognition complex subunit 6 (ORC6). Traditionally, RB pocket proteins were viewed primarily as E2F transcriptional repressors, but a paradigm shift is underway: RB family proteins are now recognized to directly control chromatin-associated activities, expanding their role beyond E2F regulation (Dick et al. Nat Rev Cancer 2018 PMID: 29692417; Sanidas et al. Mol Cell 2022. PMID: 35981542; Ding et al. Nat Commun 2022 PMID: 36274096).

Our findings, based on rapid dual protein depletion and genome-wide origin activity mapping, challenge the established view by showing that origin licensing occurs continuously throughout G1 phase, with no fixed timing for specific origins. We find CDK4/6-mediated RB phosphorylation to govern the efficiency of ORC6 and MCM loading, and all three RB pocket proteins contribute to the regulation of origin licensing by CDK4/6. Mechanistically, we now demonstrate that short, 8-hour pulses of CDK4/6 inhibition are sufficient to reduce MCM loading during G1 phase, and that the levels of CDC6, CDT1, and MCM proteins are not limiting for CDK4/6 inhibitors in blocking origin licensing. Nuclear CDT1 levels do not correlate with Palbociclib-induced ORC6/MCM loading defects, and overexpression of both CDC6 and CDT1 fails to restore licensing in Palbociclib-treated cells. Accordingly, the RB mutant defective in E2F binding remains potent in inhibiting licensing when CDK4/6 activity is low. Our conclusions are reinforced by additional experiments using selective CDK4/6 PROTACs and RB-proficient and RB-deficient cancer cell lines.

We thank the reviewers for their constructive suggestions, which helped us to significantly improve the manuscript. We concur with the reviewers that this broader role of RB pocket proteins in regulating DNA replication origin licensing has significant implications for both basic research and clinical applications and we hope that the reviewers agree that these additional data provide a more comprehensive view on origin licensing regulation in human cells. We apologize for the confusion around the essential role of CDK4/6 and we now elaborate in the discussion section on CDK redundancy and our data pointing towards a central and direct role of RB in coordinating replication licensing and cell cycle commitment. To address these changes and highlight the chemical genetics tools and implications beyond CDK regulation, we have updated the study title: “Temporal control of human DNA replication licensing by CDK4/6-RB signalling and rapid protein depletion”.

Below we will address all individual points raised by the reviewers.

Point-by-point response:

REVIEWER COMMENTS

Reviewer #1 (Remarks to the Author):

This manuscript investigates the role of CDK4/6 activity in licensing origins in G1 phase. The main concern is the overall premise of the paper since it is well established from gene knockout studies (CDK4/6 and cyclin Ds), as well as from many previous CDK4/6 inhibitor studies, that mammalian cells can license origins and replicate DNA if they lack CDK4/6 activity or have CDK4/6 inhibited. In most human non-transformed and cancer cells, CDK4/6 inhibition generally delays but does not fully prevent origin licensing and DNA replication and proliferation. As an added comment, it is well established that E2F must induce both CDC6 and CDT1 to trigger licensing, and they do not show whether this dual induction is sufficient to trigger licensing (they only induce CDC6). Moreover, they did not identify what the proposed CDK4/6 target is that could explain a potential role of CDK4/6 (in addition to its role of regulating Rb). However, the role of CDK4/6-regulated Rb phosphorylation in activating E2F is well established and could explain the lower licensing. Notably, their own, as well as many other people's Rb null data, shows that E2F activation is sufficient to license origins without a need for CDK4/6 activity. Also, Cyclin E-CDK2 activation and loss of Rb are main CDK4/6 relapse mechanisms in various cancers; both these pathways increase E2F activity when CDK4/6 is inhibited – which is consistent with E2F activation and not CDK4/6 activity directly being responsible for inducing CDC6 and CDT1 and triggering origin licensing.

1. Logical inconsistencies:

a. The authors make the claim that CDK4/6 activity in G1 is required to license origins of replication in G1. However, it is well established that cells can proliferate without any CDK4/6 activity, and origin licensing is required for proper cell cycle progression. Further, their own data in Figures 3e,f show that in the presence of Palbociclib and Mimosine, when the RBs are knocked out, cells can still load MCMs onto DNA, again arguing that CDK4/6 activity is not required for origin licensing but E2F is.

- We thank reviewer #1 for the careful evaluation of our manuscript and we agree that the requirement of CDK4/6 activity for origin licensing is not absolute. Indeed, in the absence of RB pocket proteins, MCM loading can occur independent of CDK4/6 activity (as indeed shown in Fig3) and thus CDK4/6 activity is not needed for origin licensing *per se*. We thus agree the statements should be focused on RB instead. We would like to emphasise, however, that CDK4/6 activity is a fundamental and clinically relevant regulator of RB activity, and in normal (RB-proficient) conditions origin licensing is impaired upon CDK4/6 activity loss. We have verified this phenomenon using multiple cell types (incl. RPE1, BJ, HBEC and U2OS) and by various selective CDK4/6 inhibitors and PROTACs. To better reflect these observations (and address similar suggestions of reviewers #2 and #3) we now updated the title and abstract and indicated that the CDK4/6-RB axis controls origin licensing.

- On that same note, we would like to be careful equating RB activity with E2F function, given the recent paradigm shift stressing that RB pocket proteins are not merely E2F controllers but have several E2F-independent functions that extend beyond cell cycle control (Dick et al. Nat Rev Cancer 2018 PMID: 29692417; Sanidas et al. Mol Cell 2022. PMID: 35981542; Ding et al. Nat Commun 2022 PMID: 36274096). While origin licensing *in vitro* formally does not require RB or E2F, we believe the relevance of these factors in controlling DNA replication commitment in human cells deserves attention (incl. the genome-wide and time-resolved analysis presented here). To test whether RB can restrict origin licensing in an E2F-independent manner, we used a previously identified separation-of-function RB mutant (661W) that diminishes E2F binding to undetectable levels yet allows other RB functions (Sellers et al Genes Dev 1998 PMC316399). Transient complementation of RB1^{-/-} cells with RB^{wt} or RB^{661W} revealed that RB indeed can inhibit MCM loading irrespective of its E2F-binding capability (Fig S8 and below). While ectopic expression of RB^{wt} was higher than RB^{661W}, the latter was equally potent in inhibiting licensing, suggesting that the lack of E2F binding enhanced (in contrast to being required for) RB's ability to block MCM loading. Interestingly, the interaction between hypo-phosphorylated RB and ORC1 has been shown to be competitive with the binding of RB to E2F1 (Mendoza-Maldonado et al. PLoS One. 2010 PMID: 21085491).

These single-cell data also confirm our biochemistry results (Fig. 3D-E) demonstrating that origin licensing is suboptimal in Palbociclib-treated RB1^{-/-} cells (due to the redundant activities of RBL1 and RBL2). While the latter has relevant clinical ramifications, the shared workload of RB1, RBL1 and RBL2 also explains why re-introducing RB^{wt} or RB^{661W} shows a partial yet highly significant effect on chromatin-bound MCM levels. We have included these data (Fig S8), along with other related data, in a separate results paragraph devoted to the role of E2F in licensing control.

b. Since all mammalian licensing factors (especially Cdt1 and CDC6) are highly regulated by E2F, isn't a simpler explanation consistent with the literature and their data that origin licensing is triggered by E2F activation? This likely also requires that APC/CCdh1 is active to prevent geminin accumulation (which blocks Cdt1) but APC/CCdh1 is generally active until G1/S.

- While we indeed find CDK4/6 activity to promote CDC6 expression, CDC6 protein levels are not limiting in this context, demonstrating that additional layers of regulation exist (Fig S7B). We now extended our analyses and find that nuclear CDT1 expression levels do not correlate with Palbociclib-induced licensing defects and simultaneous CDC6 and CDT1 overexpression does not restore MCM6 loading in Palbociclib treated HBEC cells (Fig S7E and below). The abovementioned RB^{661W} complementation study further supports a model where RB controls MCM loading beyond E2F regulation (Fig S8).

c. Since licensing requires E2F activation to make CDC6 and Cdt1, it seems likely that the lack of licensing is the result of delayed origin licensing in CDK4/6 inhibited cells.

In this regard, the timing when they are evaluating origin licensing with Palbociclib treatment is likely not long enough time to allow the CDK4/6-independent pathways to be increasing E2F activity (as previously shown cells with inhibited CDK4/6 are entering the cell cycle more slowly by using gradual CDK2 activation but only after some 24 hrs).

- We indeed find short-term (8-18h) CDK4/6 inhibition to reduce MCM loading, which we believe argues for direct regulation of origin licensing dynamics in cycling cells (Fig S4E, S5D and below). We wish to avoid long-term CDK4/6 inhibition for multiple days as such treatments lead to confounding secondary effects due to osmotic stress (Crozier et al. Mol Cell 2023) and MCM protein instability (Figure S4 and Crozier et al. EMBO J 2022).

- Notably, our data does not dismiss a supporting role for E2F activity in origin licensing. The Meyer lab recently identified an intermediate Rb-E2F activity state with unknown function (*Kanagaya et al. Nature 2024*), which (in addition to the speculated cell cycle plasticity or DNA repair functions) could allow origin licensing before S-phase commitment. Future studies are needed to determine how these findings based on quiescent cells re-entering the cell cycle relate to licensing dynamics in cycling cells.

d. In Figure S5b, they only overexpress CDC6, but this experiment is not enough to conclude that CDK4/6 activity is needed since CDC6 is only part of the known trigger. This conclusion can only be reached by overexpressing both Cdt1 and CDC6 since both are required for origin licensing. Also, there might potentially be additional E2F-regulated factors needed for licensing (e.g. some MCMs are E2F regulated). Again, the existing evidence shows that E2F induction of CDC6 and CDT1, but not CDK4/6 activity itself, are minimally needed as triggers for origin licensing. CDK4/6 is important indirectly, by activating E2F.

- As suggested by the reviewer, we provide additional data that simultaneous expression of CDC6 and CDT1 does not rescue the licensing defect in CDK4/6 inhibited cells (Figure S7 and comments to point 1a and 1b). Furthermore, although MCM2 and MCM6 are E2F targets (Ohtani et al. *Oncogene*. 1999), we find that short-term (8h or 18h) Palbociclib treatments impair MCM loading without causing significant loss of nuclear MCM2 and MCM6 protein levels (Figure 3C and Figure S5D), again arguing that the E2F-driven expression of these proteins is not limiting in timeframes where pocket proteins block origin licensing. We elaborate on these findings in the discussion section of the revised manuscript.

- We wish to highlight that our findings are in accordance with early observations supporting origin licensing regulation beyond E2F-driven transcriptional control. For instance, the levels of chromatin-bound MCM remain constant despite ten hours transcription inhibition by α -amanitin in K562 lymphoblast cells (Liu et al. Genome Biology 2021 PMID: 34108027), while in CHO cells, MCM loading coincides with Rb hyperphosphorylation in late-G1 phase, when mRNA suppression causes no adverse effects on MCM loading dynamics (Mukherjee et al. PLoS One 2009 PMID: 19421323). Moreover, cyclin D1-dependent kinases can effectively promote dissociation of RB-MCM7 complexes *in vitro* and promote the removal of RB from chromatin *in situ* (Gladden et al. JBC 2003 PMID: 12519773).

2. Novelty:

a. Overall, the authors do not show sufficient novel mechanisms what CDK4/6 might be doing in G1. The delay in both cell cycle entry and G1 progression with CDK4/6 inhibition has already been extensively described. As they have mentioned, the requirements of CDC6 and Cdt1 in origin licensing have also been extensively characterized.

- This is the first report that directly relates origin licensing perturbation in G1 phase to genome-wide origin activity in early S phase. As acknowledged by reviewer #3 we “*reveal some very novel observations, that are potentially of considerable interest: most importantly and surprisingly, origin licensing occurs throughout G1-phase, in contrast to the prevailing dogma that they occur at a specific point in early G1 in a window of low CDK activity; different specific origins do not have a particular timing of licensing in G1; and origin licensing may at least partially depend on CDK4/6-mediated RB phosphorylation*”. We now also provide additional mechanistic data showing that CDK4/6 activity controls origin licensing beyond CDC6/CDT1 expression regulation (Fig S4-S9 and points above).

b. If CDK4/6 indeed plays a direct role in licensing, they do not show which sites on which of the regulators of origin licensing has to be phosphorylated (CDK4/6 does not have many targets beyond Rb). More likely is that they are using in their CDK4/6i experiments conditions where E2F activity is still low or delayed in G1, explaining why they do not see origin licensing in their CDK4/6 inhibited conditions since E2F activation is delayed and not due to a new regulatory mechanism.

- Given that CDK4/6 phosphorylate several nuclear targets in G1 phase beyond RB, incl. DNA topology, nucleolar and splicing regulators (Kaulich Sci Rep. 2021 PMC8290049), we indeed wondered what the critical CDK4/6 target was. Our data shows that loss of the three RB pocket proteins completely alleviates the origin licensing defect seen upon CDK4/6 inhibition, demonstrating that RB pocket proteins are the critical substrates for licensing control (Figure 3E and 3F). These data are in line with previously reports showing both CDK4 and CDK6 phosphorylate RB pocket proteins (Anders et al. Cancer Cell 2012 PMID: 22094256), all three RB proteins bind MCM7 (Sternier et al. Mol Cell Biol 1998 PMC110654) and *in vitro* CDK4/6 activity can disrupt RB-MCM7 interactions (Gladden et al. JBC 2003 PMID: 12519773). Our experiments

indicate that RB inhibits MCM loading irrespective of its E2F-binding capacity (Fig. S8) and that overexpression of the RB-interacting domain of MCM7 disrupts RB phosphorylation and origin licensing (Fig. S9). We provide additional data using selective CDK4/6 PROTACs that confirm the strong correlation between RB phosphorylation levels and MCM loading proficiency (Figure S6 and below).

3. Other points:

a. The experiments where they overexpress C-terminal MCM7 shows reduced total RB levels, which may account for the reduction in phosphorylated RB, and the reduction in MCM2 signal (Figure S6d). The reason for this difference in total RB has not been further investigated and may be relevant. For example, the decrease in total RB may cause a reduction in detectable phospho-RB. To exclude that this is the explanation, they could use the ratio phospho-Rb over total Rb.

- We thank reviewer #1 for pointing this out and indeed we find RB phosphorylation to correlate with total RB levels, which agrees with recent reports showing that unphosphorylated RB is intrinsically unstable (Zhang et al Nature Communications 2023 PMID: 38030655). We now calculated the ratio of phospho-Rb over total Rb and updated the figure (Fig. S9 in the revised manuscript). Critically, we find MCM7-CT expression to reduce the ratio in the nucleoplasm as well as the cytoplasm fractions, supporting our conclusion that the same MCM7 peptide interferes with RB (S809/811) phosphorylation and origin licensing. Of note, Ser 809/811 is the primary target of CDK4/6, and other CDKs do not efficiently phosphorylate these sites under physiological conditions (Narasimha et al Elife 2014 PMID: 24876129) and MCM7-CT binds the N-terminal domain of RB, which is distinct from RB's pocket domain that binds E2F (Sterner et al. Mol Cell Biol 1998 PMC110654).

b. The results also are shown in RPE-1 cells, and need to be shown in other cell lines. Specifically, they should use of cancer or other cell lines that lack CDK4/6, lack RB, or are insensitive to Palbociclib treatment - this could be mechanistically informative and may change the interpretation of their data.

- We are grateful that reviewer #1 brought this up and agree it is important that our findings hold up in cancer cells. By studying origin licensing in three independent breast cancer cell lines, i.e., MCF7 (RB-proficient and sensitive to palbociclib - IC50 ~108 nM), MDA-MB231 (RB-proficient and less sensitive to palbociclib - IC50 ~227 nM), and HCC1937 (RB-deficient and palbociclib resistant), we confirmed that CDK4/6 inhibition restricts chromatin-association

of MCM2, MCM4 and ORC6 in an RB-dependent manner (Figure S5A and below). Moreover, these experiments confirmed that the decreased level of phospho-RB, but not nuclear CDT1 levels, correlate with Palbociclib-induced licensing defects (Figure S5A), further supporting our observations in RPE1 and HBEc cells (Figure S5B-C and S7E). We provide additional quantitative IF data in U2OS osteosarcoma cells confirming that low dose Palbociclib (200nM) can promptly reduce the levels of chromatin-bound MCM2 in G1 phase cells (Figure S4E). We believe these findings strengthen our manuscript and solidify our conclusions based on BJ fibroblasts and isogenic RPE1^{wt}, RPE-1^{p53 ko}, RPE-1^{RB1 ko} and RPE-1^{RB tko} cell lines.

Reviewer #2 (Remarks to the Author):

Piscitello et al. investigated the effect of CDK4/6-Cyclin D inhibitors on the replication licensing (i.e. MCM-loading onto chromatin) in G1 phase. First, the authors established an RPE1 degron cell line in which CDC6 and CDT1 could be conditionally degraded, demonstrating the MCM-loading was strongly inhibited upon the degradation of CDC6 and CDT1 (Fig. 1). Using this degron cell line, they showed the MCM-loading occurred throughout the G1 phase with no preference for specific origins to be licensed in specific G1 periods (Fig. 2). Next, the authors examined the effect of the CDK4/6 inhibitor palbociclib, revealing that it reduced the MCM-loading in G1 phase (Fig. 3a,b,c). Importantly, this licensing inhibition was rescued in cells lacking three Rb family proteins (Fig. 3d,e,f), indicating that the RB-family proteins mediate licensing inhibition caused by palbociclib. By combining palbociclib treatment with CDC6/CDT1 degradation, they demonstrated that both CDK4/6 activity and CDC6/CDT1 are simultaneously required for efficient licensing and DNA replication (Fig. 4, 5). Finally, the authors concluded that the CDK4/6-Rb axis is critical for MCM-loading in G1, thus for efficient DNA replication in S phase.

The study is carefully executed and presents high-quality data. While some results, such as mitotic cells without DNA replication (Fig. 1e and S1c) and the exploration for specific replication origins licensed at differential G1 time periods (Fig. 2 and S3), may not directly support the central conclusion, they offer valuable insights to the replication and cell-cycle research fields.

- We thank reviewer #2 for the constructive and positive comments about our work and for pointing out the added value of the presented technologies and our observations beyond CDK4/6 regulation.

The key finding is that the CDK4/6-Cyclin D/Rb pathway promotes the MCM-loading in G1 phase. CDK4/6-Cyclin D is known to phosphorylate Rb in G1 phase, releasing the E2F transcription activator from Rb and inducing the transcription of licensing factors, such as CDC6, CDT1 and MCM5/6 (Yan et al. PNAS, 1998; Ohtani et al. Oncogene, 1998; Hatenoer et al. MCB, 1998; Yoshida and Inoue, Oncogene, 2004; Ohtani et al. Oncogene, 1999). However, it remains unclear to me whether MCM-loading is mediated solely through E2F-dependent transcription or if CDK4/6-Cyclin D/Rb has a direct role in this process (Fig. 5d,e). The data supporting a direct role are Fig. 3c and Fig. S5, which show the chromatin-bound MCM2 levels were reduced despite the total nuclear MCM levels being maintained after palbociclib treatment for 18 h (Fig. 3c) and palbociclib treatment reduced MCM-loading in cells CDC6 was overproduced (Fig. S5). To strengthen the conclusion that the CDK4/6-Rb pathway has a direct role in promoting MCM-loading in G1 phase, the authors should assess the total and chromatin-bound levels of CDC6, CDT1 and MCMs by Western blotting and flow cytometry under the conditions presented in Fig. 3a, d, e, and f. This analysis would help distinguish between transcriptional and direct regulatory roles of the CDK4/6-Cyclin D/Rb axis in licensing. Revise the model presented in Fig. 5d,e.

- We agree that the manuscript benefits from additional data on how CDK4/6 promote origin licensing, and to which extent this function relies on E2F-dependent CDC6 and CDT1 expression. We have now included additional Western blot data assessing nuclear CDC6 and CDT1 levels in normal RPE1 cells and three breast cancer cell lines (Figure S5A-C and below). These experiments revealed that nuclear CDC6 levels correlate with Palbociclib-induced licensing defects (as perhaps expected for an E2F target gene), but importantly, our causal data showed that CDC6 expression levels are not limiting in this context (Figure S7B-C). Moreover, palbociclib reduces CDC6 levels also in cells lacking RB (image below; pink arrows), which contrasts palbociclib's (strictly RB-dependent) effect on origin licensing. These observations imply that i) the loss of CDC6 might reflect protein instability rather than RB/E2F-controlled transcription, and ii) the role of CDK4/6 activity in CDC6 expression can be uncoupled from its role in MCM/ORC6 loading. Although beyond the scope of this article, we believe the effect of palbociclib on CDC6 expression might be related to CDK-mediated protection against proteolysis (Mailand et al. Cell 2005 PMID: 16153703). In this light it is also good to note that unbound CDC6 is susceptible to degradation in G1 phase (Clijsters et al. J Cell Sci 2014) and that we find proteasome inhibition in Mimosine+Palbociclib

arrested cells to stabilize/elevate CDC6 and CDT1 levels without restoring origin licensing (Figure S4D).

- Could E2F-driven expression of CDT1 be the limiting factor? We believe not. Our data from RPE1, MCF7, MDA-MB231 cells indicate that Palbociclib can block origin licensing without reducing nuclear CDT1 levels (Fig. S5A-C and image below; red boxes). Importantly, origin licensing was not restored when we elevate the expression of both CDC6 and CDT1 in Palbociclib-treated HBEC cells, demonstrating that CDC6 and CDT1 levels are not limiting in this context (Fig. S7E).

- Could recruitment of CDC6 or CDT1 to chromatin be impaired by CDK4/6 inhibition? This is a valid and interesting question, but also technically challenging. The low expression levels of CDC6 and the highly dynamic and transient nature of CDT1's chromatin interactions make them hard to detect in bulk cell biochemistry assays. For CDC6 we resolved this by exploiting the HBEC CDC6-Tet-ON model, which boosts CDC6 levels and thus assay sensitivity. We found CDK4/6 inhibition not to change the levels of chromatin-bound CDC6 (Fig. S7D), while significantly reducing chromatin-bound MCM2 (Fig. S7B). For CDT1 we first aimed to detect it using our established cell fractionation and Western blot assays, but we often detected very weak and/or multiple bands in the chromatin fractions, while we detected a strong distinct band in the nucleoplasm samples (image below; top blots). Similarly, when we assessed CDT1 levels by QIBC, we readily detected nuclear CDT1 intensities in G1 cells, but nearly all this signal was lost after pre-extraction, indicating that the vast majority of CDT1 was not chromatin-bound (image below; bottom scatter plots). The total nuclear levels of CDT1 showed a clear cell cycle dependent pattern, with the G1 signal dropping sharply in S-phase (in line with its active degradation by Cul4-DDB1-Cdt2). In contrast, the chromatin-bound CDT1 levels showed minor cell cycle dependent variations and G1-phase levels were very close to baseline (i.e. S-phase levels). While our western blot and QIBC data indicates that these residual chromatin-bound CDT1 signals do not change significantly upon Palbociclib treatment, we prefer not to make any strong

claims on such low signals. As we felt uncertain whether these residual signals reflect the dynamic pool of CDT1 that loads MCMs, we decided not to include these results into the revised manuscript.

- Based on these observations and similar IF data on other E2F target genes such as MCM6 (Figure S5D) as well as the experiments expressing the MCM7-CT peptide (Figure S9) and mutant RB^{661W} (Figure S8), we postulate that RB controls MCM and ORC6 loading beyond E2F-dependent mechanisms. We have updated the discussion and model figures accordingly (Fig. 5D, 5E).

[REDACTED]

Minor points

- In Fig. S7b, the authors showed that p53 positive cells did not enter mitosis when licensing was inhibited by palbociclib and ORC inhibitor RL5a. I did not see any explanation related to this result. Can you discuss why p53 positive and negative cells showed differential phenotypes?

- We now elaborate on these results and the p53-dependent origin licensing checkpoint (Nevis et al Cell Cycle 2009 PMID: 19440053) in the discussion section.

- In Fig. 3d,e,f, it is not clear how long the cells were treated with mimosine

and palbociclib. The experimental condition should be indicated in the figure legend. These are key data in this paper.

- In all cases the cells were treated with Mimosine for 22 hours and Palbociclib for 26 hours. We now added these details to the figure legend.
- The authors should refer to the supplementary figure panels in the main text rather than just referring to Fig. S1 or S2. It is difficult to look at the supplementary figure while I read the main text.
- Thank you for this suggestion to improve readability. We have updated the figure references accordingly.

Reviewer #3 (Remarks to the Author):

Review of Sesenko Piscitello et al

This is a very interesting study on important and inter-related unresolved questions about the mammalian cell cycle: whether origin licensing occurs at a specific point in G1 phase, and whether it depends on CDK4/6 activity. The authors use an innovative combination of degron-tagging of two licensing proteins to chemically control replication licensing at different stages post-mitosis and block-release experiments with CDK4/6 inhibitors and/or mimosine, a drug that prevents S-phase onset, followed by analysis of replication using single-cell techniques with genomic resolution, as well as analysis of chromatin loading of replication factors. They reveal some very novel observations, that are potentially of considerable interest: most importantly and surprisingly, origin licensing occurs throughout G1-phase, in contrast to the prevailing dogma that they occur at a specific point in early G1 in a window of low CDK activity; different specific origins do not have a particular timing of licensing in G1; and origin licensing may at least partially depend on CDK4/6-mediated RB phosphorylation, at least in this system. The latter is perhaps the weakest of the conclusions (see below), and so it is a shame that the title of the paper focused on it. Using genetic knockout of paralogues, they further note a partial redundancy of RB family proteins in this control and show evidence of a potential mechanism by which RB family protein phosphorylation may work. All of these are very useful observations that could potentially contribute to establishing a new paradigm about this critical phase of the cell cycle.

- We thank reviewer #3 for the shared enthusiasm and helpful suggestions on how to better highlight the added value of the presented technologies and findings beyond CDK4/6 regulation. Based on these and other reviewer's comments we propose a new study title: "Temporal control of human DNA replication licensing by CDK4/6-RB signalling and rapid protein depletion".

I feel, however, that some revisions are required. First, I think the CDK4/6i experiments have been slightly misinterpreted because of the way they were performed. This hinges around the claim made in the title: "Licensing of human DNA replication origins requires CDK4/6 activity in G1 phase". On an aside, by talking uniquely about human cells, this sidesteps the uncomfortable question about why the cell cycle, and DNA replication, is unaffected by genetic knockout of both CDK4 and

CDK6 in mice. This work was neither mentioned nor cited, whereas it seems unlikely that mouse cells do anything very differently to human cells, and I think there are ways of reconciling the data of the current paper with the mouse data (see below).

- We fully agree with the interpretation that lack of CDK4/6 proteins (due to genetic knockout) likely allows redundant CDKs to take over, inhibit RB and sustain viability. Our data supports the view that in situations where RB pocket proteins are inactivated (either by hyperphosphorylation or genetic depletion) the role of CDK4/6 activity in origin licensing can be bypassed (e.g. Figure 3F). The reason for us to highlight human cell biology is not to sidestep the solid mouse work but to highlight the possible ramifications for human medicine. Numerous clinical trials and pre-clinical studies are underway that exploit CDK4/6 targeting, including the development of selective kinase inhibitors and PROTACs. We now provide additional data showing that PROTACs targeting CDK4/6 protein stability also arrest human cells in an under-licensed state (Fig S6 and point 2b; Reviewer#1). While the extent of the MCM2/MCM6 loading defect in RPE1 cells exposed to Palbociclib or combined CDK4 and CDK6 PROTAC treatment was undistinguishable (Fig. S6C-D), we observed some notable differences in the cell population. First, while the PROTACs reduced CDK4/6 protein levels, the catalytic inhibitor Palbociclib increased CDK4/6 protein levels (Fig. S6A). Second, RB phosphorylation was more potently inhibited by Palbociclib than combined CDK4 and CDK6 PROTAC treatment (despite the latter being added at significantly higher concentrations to compensate for any activity loss towards the catalytic site (Figure S6A). For instance, we added 2.5-fold more BSJ-03-123 since in vitro activities towards CDK4/6 of BSJ-03-123 is 42/9 nM (IC50) and Palbociclib is 14/15nM (IC50). These data support a model where i) both CDK4 and CDK6 promote cell cycle progression and MCM loading in human cells, and ii) redundant CDK1/2 activities exist that can phosphorylate RB in the absence of CDK4/6, yet those CDKs might be poor promoters of origin licensing. The latter conundrum provides a *raison d'être* for separate CDK families: one dedicated to promoting RB phosphorylation and origin licensing (i.e. CDK4/6), and one to RB hyperphosphorylation, S-phase entry and origin firing (i.e. CDK1/2).
- We have included these PROTAC studies in the result section (Western blot and IF data) and added a discussion on CDK redundancy and the mouse viability data.

Equally importantly, I would suggest that it is unlikely that CDK4 and CDK6 do anything specific in G1, and this also appears to have been overlooked. Work from former members of Tobias Meyer's team (Steven Cappell and Sabrina Spencer) has shown that CDK4 and CDK6 operate throughout the cell cycle; in particular, Sabrina Spencer's team has shown evidence that pre-mitotic CDK4/6 activity affects the following G1. Her original data published as a postdoc in the Meyer lab also do not fit with the traditional but probably outdated model (see reviews by Steven Cooper) of the "restriction point" in G1, since serum withdrawal immediately after anaphase does not prevent cell cycle progression. To come back to the current paper, I think it is very likely that if CDK4/6 had been inhibited specifically prior to mitosis, or at any point after beginning of the previous S-phase, they would also see a defective G1-S phase progression in the following cell cycle due to insufficient origin licensing. Thus, I think it is premature to state that origin licensing requires CDK4/6 activity in G1.

That it requires CDK4/6 activity at some point in the cell cycle, however, seems likely, and some experiments would easily address this.

- We are grateful for these considerations, and we agree that CDK4/6 activity modulation (shortly) prior to mitosis could well influence origin licensing at mitotic exit. It was in fact recent literature (e.g. Cornwell et al. Nature 2023) that moved us to include “G1 phase” in the title, since these latest models imply a sustained role for CDK4/6 in all cell cycle stages. We wished to focus/constrain our conclusion on G1 phase because our time-resolved EdUseq data subjected cells to CDK4/6 inhibition specifically during early or late G1 phase (Figure 5). Using a synchronized high-content IF approach, we now directly compared the impact of CDK4/6 inhibition before or after mitotic exit and assessed chromatin-bound MCM levels in single-cells (Fig. S11 and below). First, we show that 8 hours of palbociclib treatment in cycling RPE1 cells is sufficient to stall MCM loading in the G1 phase population (Fig. S11A-C). Since this was an asynchronous population, some of these G1 cells would have been in G1 during the entire 8-hour treatment, while other cells entered G1 phase only in the last hour and thus were exposed to palbociclib as they traversed G2/M phase. To study how the relative timing of CDK4/6 inhibition controls origin licensing efficacy, we used a synchronized setup based on mitotic-shake-off in which cells were exposed to palbociclib during the 3-hour mitotic arrest, 8-hours after mitotic release or both. These experiments demonstrated that palbociclib treatment in G1 phase (i.e. post mitotic release) was sufficient to restrain origin licensing dynamics (Fig. S11), which supports the observed epistasis between CDK4/6 inhibition and CDC6/CDT1 loss in G1 phase (Fig 5; middle centre panel). These data also showed that the licensing inhibition was most effective when palbociclib was added during the mitotic arrest and kept present during G1 phase entry (Fig. S11), implying that unphosphorylated RB is able to interfere with MCM loading as soon as cells exit mitosis. Because of these results (and also to highlight our chemical genetic approach that could be useful beyond CDK-RB research) we propose to change the study title to: “Temporal control of human DNA replication licensing by CDK4/6-RB signalling and rapid protein depletion”.

Second, the conclusion of a specific block of licensing by CDK4/6 inhibition is not completely convincing. While in Fig 3a, the reduction in chromatin-bound MCM is obvious, CDC45 loading (which is downstream of MCM) is only slightly reduced, and is not further reduced by a combination of CDK4/6i that give a tighter block over RB phosphorylation. DNA replication was apparently not analysed in this experiment, and so it cannot be concluded that CDK4/6i block licensing. It is also surprising that chromatin loading of the licensing factors CDC6 and CDT1 was not assessed in these experiments.

We thank reviewer#3 for the careful considerations and have updated the result section to include additional CDC6 and CDT1 expression studies (Fig S5A-C and comments to reviewer #2). Regarding the CDC45 data interpretation, we like to highlight that the firing factor CDC45 is recruited to chromatin in S-phase and thus the chromatin-bound CDC45 signal in bulk cell populations (Figure 3a) likely originates from different cells (S-phase cells) than those showing the reduced MCM2 loading (G1 phase cells). We used low dose and relative short-term CDK4/6 inhibitor treatments to minimize crosstalk with CDK1/2 activities and prevent adverse side effects on MCM protein stability (Crozier et al. EMBO J 2022). As a result, G1 synchronization is not 100% complete (see Figure 3B for single cell data). We interpret that the drop in CDC45 signal seen upon CDK4/6 inhibition (Figure 3a) is mainly due to cell cycle synchronisation (reduction of the S phase population) and the residual CDC45 bands originate from the unsynchronized subpopulation. This signal therefore does not reflect successful recruitment of CDC45 in the G1 arrested cells that display the MCM2/4/6 loading defects. While we believe the result on steady levels of CDC45 in the bulk assay is correct (reflecting synchronisation) and serves as a control to highlight the dynamic MCM2 changes, we also see this might confuse many readers. We thus have decided to exclude the CDC45 blot from the revised manuscript, but we are happy to include it if requested.

Indeed, the point at which the MCM loading defect occurs is not entirely clear, partly since some experimental conditions used CDK4/6i alone (eg Fig 3a lanes 1-3 and 5) and assessed some factors, while others used CDK4/6i on top of mimosine (eg Fig 3b, d-f, Fig S5b-d) and assessed an additional factor, namely, ORC6, whose chromatin loading is required for CDC6/CDT1 loading. It is striking that ORC6 levels decrease upon CDK4/6i treatment.

- Recent studies indeed suggest that ORC6 deficiency hampers MCM loading *in vitro* (Yang et al Nature 2024 PMID: 39604729) and in RPE1 cells (Hayashi-Takanaka et al BioRxiv 2024.01.30.577900) but surprisingly not in U2OS cells (Lin et al PNAS 2022 PMID: 35622890). We now provide additional data demonstrating that chromatin-bound ORC6 decreases upon palbociclib treatment in MCF7 and MDA-MB231 cells, but not in RB-deficient HCC1937 cells (Fig. S5A) substantiating our previous observations in RPE1 cells indicating that this ORC6 phenotype required RB pocket proteins (Fig. 3D-E). To our knowledge, no study has yet connected CDK4/6-RB signalling to ORC6 function and we thus believe these data sets provide valuable new insights into

the regulation of origin licensing, thereby also boosting of novelty of our manuscript.

The experiments performed to address the stage of the block, and shown in Figure 4, were interpreted as showing that CDK4/6i inhibit licensing upstream of CDC6/CDT1 function, since loss of the latter two proteins upon CDK4/6i release did not allow DNA replication, whereas the control could still replicate. However, it is an experiment in asynchronous cells, in which CDK4/6i will act at all stages of the cell cycle, and it is possible that as a consequence some cells have not passed mitosis, while other cells having passed mitosis have not synthesised sufficient factors required for licensing (ORC, CDC6, CDT1). This, and their loading onto chromatin, should be assessed.

- While the experiment setup of Figure 4 starts with asynchronous RPE1 CDC6^d CDT1^d cells, the majority of cells (~80%) was arrested in G1 cells after 26 hours of palbociclib treatment (data below). These cells lack p53 and thus synchronization is less efficient compared to RPE1^{wt}, but these responses are very similar to RPE1^{p53-/-} (Fig. 3A and 3C). While we agree that a minority of cells (~20%) might not have passed mitosis upon palbociclib release, we would like to emphasize the severity of the observed DNA replication defects when these cells were released in S phase cells with degraded CDC6 and CDT1, i.e., EdU was virtually absent in early-S phase cells or at least 4-fold lower than controls in late-S phase (Fig 3B-C). We believe these phenotypes cannot be explained by a minor subpopulation and thus it is most parsimonious that CDK4/6-inhibited G1 phase cells remain sensitive to CDC6/CDT1 loss. We conclude that the palbociclib-induced block acts upstream or epistatic to CDC6/CDT1 function, as we expect a downstream late-stage block to be insensitive to CDC6/CDT1 loss. We demonstrate this alternative scenario empirically via a Mimosine-induced replication block, which is thought to occur at the transition from pre-RC to pre-IC and indeed is insensitive to CDC6/CDT1 loss (Fig. S10).
- Although we have not determined whether palbociclib impairs licensing via E2F-driven protein synthesis in this very setting with RPE1 CDC6^d CDT1^d cells, we provide additional data arguing that palbociclib's mode-of-action does not require CDC6, CDT1, ORC1 and MCM protein levels to be limiting (Fig S4-S8).

The complementary experiment has been performed: blocking cells by degrading CDC6/CDT1, and then releasing into palbociclib (Fig 5c), but unfortunately licensing was not analysed in this experiment. If it had been done, it would allow to rule out that CDK4/6i prevent MCM loading to already-bound CDC6/CDT1. DNA replication is still inhibited, as expected, since origin firing

and thus DNA replication require the triggering of licensed origins by other CDK (CDK1 or CDK2), and CDK4/6i inhibits the expression of cyclins required to activate them. As it stands, it cannot be concluded from Fig 5 experiments that CDK4/6 activity is required at the time of origin licensing, although these experiments are very interesting since they suggest that CDK4/6 is not required at a particular “restriction point” but acts throughout G1 (if not the entire cell cycle).

- We agree that the classical model where CDK4/6 acts only at the restriction point in late G1 to release E2F needs to be revisited. Our data is not mutually exclusive with a model where CDK4/6 are active throughout the entire cell cycle and in which E2F-driven expression of licensing factors in S phase prepares the cell for origin licensing in the next G1 phase.
- We provide additional high-content IF data supporting our claim that CDK4/6 activity promotes MCM loading in early G1 phase (Fig. S11). These data explain why we detect less origin activity by EdUseq – especially when followed by CDC6/CDT1 degradation (Fig. 5, upper right and mid-right panel).
- We agree that sustained loss of E2F transcription will prevent the accumulation of S-phase cyclins and thus on the long run diminish origin firing potential, however, it is unclear to us how such a defect in firing would explain the complete loss of origin activity in cells depleted of CDC6/CDT1 in early G1 and released in palbociclib in late G1 (Fig. 5; bottom middle panel). If firing would be prevented by the 10-hour pulse of palbociclib in late G1 phase, would one not expect an equally strong DNA replication defect in the two conditions above (Fig. 5; centre panel and top middle panel)? Similarly, we would anticipate licensing defects and firing defects to show additive effects (as DNA replication capacity depends on both loading and activation of MCM complexes). Our EdUseq experiments, however, show epistatic relationships between CDC6/CDT1 degradation and CDK4/6 inhibition (Fig. 5; center and left middle panel), implying that the latter causes a similar (licensing) defect.

How can the interpretation of the authors be reconciled with the observation that CDK4/6 knockout in mice does not affect DNA replication? Likely because upon genetic knockout of CDK4/6, or even CDK2/3/4/6 (Santamaria et al., 2007), CDK1 steps in to rescue RB phosphorylation. CDK4/6 inhibition is qualitatively different in that cyclin D is still present and can still interact with RB, but can no longer phosphorylate it. This should also be discussed, and it might also necessitate a modification of the title: CDK4/6 are not necessarily required for licensing, because in their absence CDK1/2 can do it. Instead, one could say that CDK4/6 promote licensing, or, an alternative explanation that could perhaps fit better with the data in the paper, they promote the acquisition of a licensing-competent state by allowing synthesis of the licensing factors themselves (which would be novel and interesting). This of course requires confirmation as described above. Notwithstanding these criticisms, I think the findings on origin control and CDK4/6 action throughout G1 (Figures 1, 2, 5) are extremely interesting and important and would deserve a very visible publication. In conclusion, I think this is an excellent and important study but requires some work before publication.

- We thank the reviewer for the title suggestions and agree with the reviewer’s interpretation of the mouse knockout phenotypes. Given the key role for the

pocket proteins (regardless of the upstream CDK), we wish to include RB in the title. Since CDK4/6 activity promotes licensing but RB does the opposite, our revised title states that the CDK4/6-RB axis controls timely licensing. We also updated the abstract accordingly.

Response to the reviewers concerning revised manuscript NCOMMS-24-81972B

We thank the reviewers for their critical evaluation and constructive feedback. Below we will address all individual points raised by the reviewers.

Point-by-point response:

REVIEWERS' COMMENTS

Reviewer #1 (Remarks to the Author):

The authors have addressed my main issues.

Thank you for your positive comment and for helping us improve our manuscript

Reviewer #2 (Remarks to the Author):

The authors added new data supporting the evidence that CDJ4/6-Cyclin D has a direct role in origin licensing, mediated by the RB family proteins. They have clearly addressed my previous concerns.

To make the final version, the authors should double-check the order of Figures S5 to S7, as the text on pages 7-8 does not align with these figures. Probably, Figure S5 should be Figure S7. Additionally, Figures 4d and S10 are missing scale bars.

Thank you for your valuable comments and thorough review. We have revisited the order of the extended data figures and adjusted the references in the text accordingly. We also added the missing scale bars.

Reviewer #3 (Remarks to the Author):

Lemmens, Halazonetis and colleagues have taken into account comments of all reviewers and performed useful additional experiments to address the most important points raised. This is indeed an improved manuscript since it now shows strong evidence that the phenotypes of loss of RB/RBL activity on licensing are partly direct, ie independent of E2F-mediated transcription, which was a major criticism of reviewers 1 and 2. This can of course still be debated but I agree that the balance of evidence points to direct effects that are at least independent of CDC6 and CDT1 levels. The authors also now address, mostly by discussion, the model whereby CDK4/6 can be compensated for by CDK1/2 in the absence of the former. I would have a different interpretation: namely that CDK4/6 promote CDK1/2 activity (not necessarily just by promoting E2F transcription), and either activity can promote licensing and/or firing, with nothing special about CDK4/6 other than the fact that it is downstream of growth regulating pathways and upstream of CDK1/2. The fact remains that the authors have not formally demonstrated that CDK4/6 do not also do something downstream of CDC6/CDT1 chromatin loading/licensing, and, since they have only tested CDK4/6

inhibition, they have also not formally excluded that CDK1/2 inhibition might have similar effects on licensing as CDK4/6 inhibition (though it will also inhibit firing). But both models are useful and further studies in the field will indicate which is correct.

Thank you for your careful review and valuable insights. We agree that it will be interesting to further investigate the interplay between CDK1/2 and CDK4/6 activities in the future. CDK1/2 and CDK4/6 have fundamentally different roles in controlling mitotic progression and origin firing, which puts CDK4/6 in a 'special' position to control origin licensing throughout G1 phase. CDK4/6 inhibitors are also better tolerated in patients than CDK1/2 inhibitors, providing opportunities to leverage replication licensing control in the clinic.

I would consider changing the title again because it has now become a bit of a non-sequitur, it mixes temporal control by the cellular biochemistry (CDK4/6/RB) and rapid protein degradation, without specifying that the latter is an experimental system and not a cellular control. It also, in my view, fails to fully capture the impact of the study. It would read better something like: "Rapid protein degradation in human cells reveals direct promotion of DNA replication licensing throughout G1 by CDK4/6-RB signalling".

We agree that the title might imply a biological connection between CDK4/6-RB signalling and rapid protein degradation, while it was intended to highlight our improved double-degron system. We therefore propose a new title:

"Temporal control of human DNA replication licensing by CDK4/6-RB signalling and chemical genetics".

With this new title, we aim to distinguish between biology and technology, while honouring the 15-word limit. We also agree with reviewers' previous comment that CDK4/6 activity might control origin licensing beyond G1 phase (including early licensing steps post anaphase or during pathological hyper-replication conditions) and thus choose not to restrict the title to G1 phase.

Finally, there are quite a few typo or English errors in the manuscript (eg lines 172, 274, 334, 348, 360, 375, 419, 455).

Thank you also for pointing out the grammar and spelling errors. We have corrected these in the revised text and performed an additional spelling check.

Once these are corrected, I do not see any reason to delay the publication of this manuscript, which makes an important contribution to understanding cell cycle control and DNA replication in particular.